
# A Maximum Entropy Production Evaporation -
# Transpiration Product for Australia
Olanrewaju Abiodun[1], Okke Batelaan[1] Huade Guan[1], and Jingfeng Wang[2]
1. National Centre for Groundwater Research and Training, College of Science and Engineering, Flinders
University, Australia
2. School of Civil and Environmental Engineering, Georgia Institute of Technology, Atlanta, USA
Corresponding author:
Olanrewaju Abiodun, College of Science and Engineering, Flinders University, Bedford Park, Australia,
lanre.abiodun@flinders.edu.au
**Abstract**
The aim of this research is to develop evaporation and transpiration products for Australia based on the maximum
entropy production model (MEP). We introduce a method into the MEP algorithm of estimating the required
model parameters over the entire Australia through the use of pedotransfer function, soil properties and remotely
sensed soil moisture data. Our algorithm calculates the evaporation and transpiration over Australia on daily
timescales at the 5 km$^2$ resolution for 2003 – 2013.
The MEP evapotranspiration (ET) estimates are validated using observed ET data from 20 Eddy Covariance (EC)
flux towers across 8 land cover types in Australia. We also compare the MEP ET at the EC flux towers with two
other ET products over Australia; MOD16 and AWRA-L products. The MEP model outperforms the MOD16 and
AWRA-L across the 20 EC flux sites, with average root mean square errors (RMSE), 8.21, 9.87 and 9.22 mm/8
days respectively. The average mean absolute error (MAE) for the MEP, MOD16 and AWRA-L are 6.21, 7.29
and 6.52 mm/8 days, the average correlations are 0.64, 0.57 and 0.61, respectively. The percentage Bias of the
MEP ET was within 20% of the observed ET at 12 of the 20 EC flux sites while the MOD16 and AWRA-L ET
were within 20% of the observed ET at 4 and 10 sites respectively. Our analysis shows that evaporation and
transpiration contribute 38% and 62%, respectively, to the total ET across the study period which includes a
significant part of the "millennium drought" period (2003 – 2009) in Australia. The data (Abiodun et al., 2019) is
available at http://dx.doi.org/10.25901/5ce795d313db8



27    .

*Keywords:* Evaporation; transpiration; Maximum Entropy Production; remote sensing
**1. Introduction**
The use of remote sensing data in existing and new methods for evapotranspiration (ET) estimation is
incontrovertibly the current and future trend of ET flux quantification on catchment, regional and continental
scales (Bhattarai et al., 2016;Zhang et al., 2016;Najmaddin et al., 2017). The use of remote sensing observations
is an unprecedented advancement in regional scale ET estimation due to its spatiotemporal flexibility and/or
economic viability (Chirouze et al., 2014;Long et al., 2014;Xiong et al., 2014;Yang et al., 2015;Bhattarai et al.,
2016). Various methods have been developed for improving ET estimates (Allen et al., 2007;Cleugh et al.,
2007;Tang et al., 2009;Mu et al., 2011;Xiong et al., 2014). However, the relative accuracy of these methods
differ across different climates, vegetation and soil types (Jia et al., 2012;Kim et al., 2012;Velpuri et al.,
2013;Bhattarai et al., 2016). The performance of the ET models depends on the parameterization of physical
processes underlying ET (Liaqat and Choi, 2017). A major challenge is to produce accurate ET estimates of
various spatial and temporal resolutions (Senay et al., 2013;Wang et al., 2016;Gaur et al., 2017) when using
remote sensing data (Kalma et al. (2008).
A remote sensing based ET model is empirical or physically-based (Xiong et al., 2014). In the past two decades,
several physically based ET models have been developed including the single source energy balance (SSEB)
(Bastiaanssen et al., 1998;Roerink et al., 2000;Allen et al., 2007) and two-source surface energy balance (TSEB)
(Kustas and Norman, 1999;Norman et al., 2003;Sun et al., 2009) models using remote sensing input data. The
SSEB models provide total ET without partitioning it into soil evaporation (E) and transpiration (T), while the
TSEB models do the partition. The TSEB models have been shown to be more accurate over partially vegetated
surfaces (Timmermans et al., 2007;Gao and Long, 2008;Choi et al., 2009). A fundamental challenge of TSEB
models is their reliance on land surface temperature (LST) and the partitioning methodology of the LST into soil
and canopy temperature components for modelling (Colaizzi et al., 2012;Yang et al., 2018). Different
techniques have been applied to partition the canopy and soil temperatures from the LST in the TSEB models
(Norman et al., 2000;Zhang et al., 2005), with varying degree of success over different vegetation types (Chavez
et al., 2009;Song et al., 2016;Diarra et al., 2017). The more pertinent challenge of the TSEB models becomes
apparent when creating high resolution regional to continental scale ET, which requires accurate LST data as the



principal input. Frequent clouds plague remotely sensed LST products such as the widely accepted Moderate
Resolution Imaging Spectroradiometer land surface temperature product (MODIS LST) (Wan et al., 2002).
The limitations of the LST dependence of the traditional TSEB models was further highlighted by Mu et al.
(2007) who found that the use of the 8-day composite of all cloud free data in the MODIS LST suite did not
produce accurate estimates of global scale evapotranspiration. The MODIS LST yielded erroneous results of
partitioned soil and canopy temperatures across various biomes, hence the development of a new algorithm is
needed for estimating soil and canopy temperatures for improving the MODIS ET product (MOD16), which is
widely accepted for comparison and validation purposes on catchment to continental scales. There are, however,
unresolved issues of accuracy (Tang et al., 2015;de Arruda Souza et al., 2018;Khan et al., 2018). With the
challenge surrounding the LST partitioning in TSEB models and the MOD16 challenges, a different perspective
to the TSEB modelling on regional scale is required.

The Maximum Entropy Production (MEP) model of ET (Wang and Bras, 2011) is a new approach to modelling
ET. The MEP model was formulated as a unique TSEB model for soil and vegetated surface where ET and the
other surface heat fluxes result from the partition of net radiation. The MEP model requires three main inputs:
surface temperature, specific humidity and net radiation. A major departure of the MEP model from the
traditional TSEB models is that the MEP model is less sensitive to temperature and more sensitive to the
moisture content of immediately above the target surface and the available energy.
Case studies have shown that the MEP ET for small catchments outperformed several other models (Nearing et
al., 2012;Yang and Wang, 2014;Shanafield et al., 2015). However, the MEP ET model is yet to be
comprehensively tested over various vegetation covers. A global product of the MEP ET at a 100 km² spatial
resolution has been produced (Huang et al., 2017). However, at this scale, individual vegetation cover type
validation and analysis is problematic. The ET data over the diverse Australian landscape at catchment to
continental scale has been produced (Guerschman et al. (2009) using MOD16 model (Mu et al., 2011) and the
Australian Water Resource Assessment Landscape (AWRA-L) model (Viney et al., 2014).
The goal of this paper is to develop a daily MEP ET product for Australia on a 0.05° spatial resolution. We have
generated the data for 2003 – 2013 for demonstration and testing of result (Abiodun et al., 2019). The skill of
the MEP ET model will be evaluated using eddy covariance tower data across various vegetation covers and



compared with the results of the MOD16 and the AWRA-L products. The evaluation period covers the
climatological highly variable "millennium drought" period (2003-2010).

**2   Method and data**

The energy balance equation over the land surface is expressed as,
$E + H + G = R_n$                              (1)
where $E, H, G$ and $R_n$ are evapotranspiration (W/m²), sensible heat (W/m²), ground heat (W/m²) and net radiation
(W/m²), respectively. The MEP ET model provides a solution of $Es, Hs$, and $G$ over non-vegetated land surface
satisfying the energy balance equation Eq. (1) (Wang and Bras (2011) for given net radiation Rn, surface
temperature T, and surface specific humidity q,
$\sigma_s = \frac{\lambda^2}{c_p R_v} \frac{q_s}{T_s^2}$ ,   $\beta(\sigma_s) = 6\left(\sqrt{1 + \frac{11}{36}\sigma_s} - 1\right)$                              (2)
$G = \frac{\beta(\sigma_s)}{\sigma_s} \frac{I_s}{I_o} H_s |H_s|^{-\frac{1}{6}}$                              (3)
$E_s = \beta(\sigma_s) H_s$                              (4)

where $\sigma_s$ (Sigma) is a dimensionless parameter characterizing the effect of (soil or canopy) surface thermal and
moisture state on the phase change of liquid water (-); $\lambda$ is the latent heat of vaporization of liquid water (J kg⁻¹);
$c_p$ is the specific heat of dry air at constant pressure (J kg⁻¹ K⁻¹); $R_v$ is the gas constant of water vapor (J kg⁻¹ K⁻¹);
$q_s$ the specific humidity at the soil or vegetation surface (kg kg⁻¹); $T_s$ is the soil or canopy surface temperatures
(K); $\beta(\sigma_s)$ is the inverse Bowen ratio (-); $I_s$ is the thermal inertia of soil (J m⁻² K⁻¹ s⁻¹ᐟ²); $I_o$ is the thermal inertia
of turbulent air (J m⁻² K⁻¹ s⁻¹ᐟ²). For vegetated land surface where $G$ is neglected, equations (2) – (4) become;

105          $E_v = \frac{R_{n\_v}}{1 + \sigma_s^{-1}}$ , $H_v = \frac{R_{n\_v}}{1 + \sigma_s}$                              (6)




where $E_v$ is the canopy transpiration and $H_v$ sensible heat flux over canopy surface satisfying energy balance
equation $R_n = E_v + H_v$.
The MEP ET algorithm calculates soil evaporation and canopy transpiration separately. Total evapotranspiration
is the sum of the two fluxes weighted by the fractional coverage of soil and canopy (Fig 1). In this paper, we apply
temporally varying vegetation fraction cover in the algorithm to partition the radiation energy for soil and canopy.

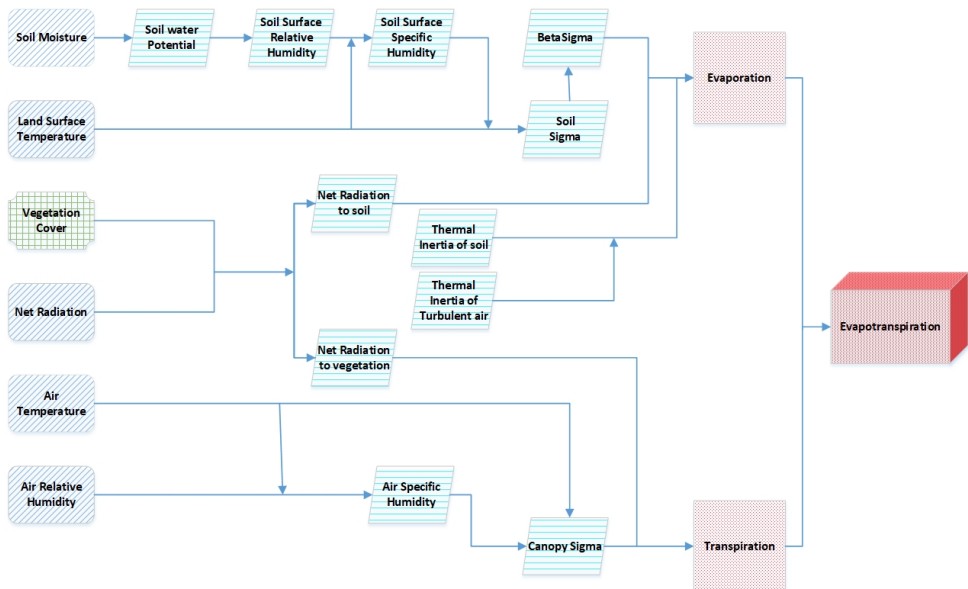


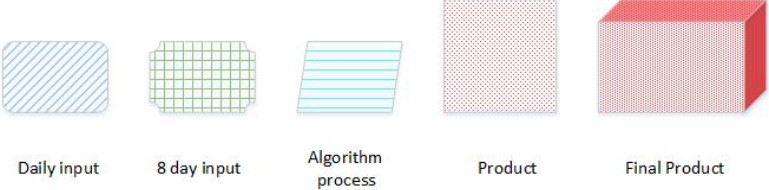


*Figure 1: Flowchart of MEP ET algorithm; BetaSigma is the inverse Bowen ratio*

**2.1   Net radiation ($R_n$)**



Daily net radiation at 0.05º spatial resolution over Australia is partitioned between soil and canopy within a grid
cell according to vegetation fraction cover. Photosynthetically active radiation (FPAR) product MOD15A2H
(Myneni et al., 2015)  is used in this study. While the MEP model is very sensitive to net radiation as a model
input with pronounced diurnal cycle, 8-day vegetation cover data were used as vegetation cover changes at
seasonal time scale. Net radiation over canopy and soil surface within a grid cell is expressed as,
$$R_{n\_v} = F_c\,R_n \;\;,\;\; R_{n\_s} = (1 - F_c)\,R_n \tag{7}$$
where, $R_{n\_v}$ is the net radiation over vegetation (W/m$^2$), $R_{n\_s}$ is the net radiation over soil (W/m$^2$), and $F_c$ is the
vegetation fraction (-).

## 2.2  Evaporation


The MEP model as in Eqs. 1, 3 and 4 provides a unique solution of E, G and H for given surface temperature ($T_s$),
soil/canopy surface specific humidity ($q_s$), and $R_{n\_s}$. The land surface temperature ($T_s$) is provided by the
MOD11C1 product (Wan, 2014) derived from the MODIS observations. The daily data for Australia from 2003
to 2013 was extracted from the global dataset. Missing $T_s$ data, due to cloud cover, were filled using the lowest
value within a month for each grid cell. The rationale is that cloud cover reduces the amount of solar radiation
reaching the land surface, hence the lowest observed $T_s$ value within a month is used.
Due to the difficulty of obtaining $q_s$ over the entire Australia, an empirical equation is used to calculate $q_s$ as a
function of soil surface relative humidity and land surface temperature. The soil surface relative humidity is
calculated from the soil surface water potential. The Hutson and Cass function (Hutson and Cass (1987) is used
for estimating soil surface water potential. The Hutson & Cass function requires two empirical coefficients
calibrated for each grid cell using two methods: the empirical equation derived in Williams et al. (1992), and the
pedotransfer functions to estimate the soil water content at wilting point (-1.5MPa) and at field capacity (-10kPa).
The water content at the wilting point and field capacity for each 0.05º grid cell, estimated from the pedotransfer
functions, are subsequently used to determine the coefficients, by applying the two-point method (Cresswell and
Paydar (1996) (see Section 2.3.1). Different pedotransfer functions for determining the wilting point and field
capacity (Minasny et al., 1999;Minasny and Mcbratney, 2002;Rab et al., 2011) (see Equations. 12 and 13 in (Rab
et al. (2011)) were selected due to their modest data requirement and relative accuracy. The pedotransfer function
combined with the two point method was preferred to the empirical equations (Williams et al. (1992) as they



yielded significantly better estimates of ET after validation with flux tower data. Soil properties as the inputs of
the pedotransfer functions and empirical equations are obtained from the Australian Soil Resource Information
System (ASRIS) (Johnston et al., 2003).
An important parameter of the MEP model is the distance above target surface for which the Monin-Obukhov
similarity theory is valid ($z$) in the formula of the thermal inertia of turbulent air above soil surface. Huang et al.
(2017) suggested that the distance above target ($z$) vary with the land cover types as shown in the look-up table
(Table 1) used in this study. $z$ for each land cover is specified for each 0.05º grid cell using the MODIS land cover
product (MOD12C1) (Mark and Damien, 2015) of the same resolution.
*Table 1: Distance above target surface (z) in (m) for Australian Land cover*

| Land Cover | Distance above target (z) in (m) |
|---|---|
| Evergreen Needleleaf Forests (ENF) | 10 |
| Evergreen Broadleaf Forests (EBF) | 10 |
| Deciduous Needleleaf Forests (DNF) | 10 |
| Deciduous Broadleaf Forests (DBF) | 10 |
| Mixed Forests (MF) | 10 |
| Closed Shrublands (CSH) | 5 |
| Open Shrublands (OSH) | 4 |
| Woody Savannas (WSA) | 8 |
| Savannas (SAV) | 7 |
| Grasslands (GRA) | 5 |
| Croplands (CRP) | 5 |
| Urban and Built up (URB) | 3 |
| Cropland/Natural Vegetation Mosaics (CRV) | 5 |


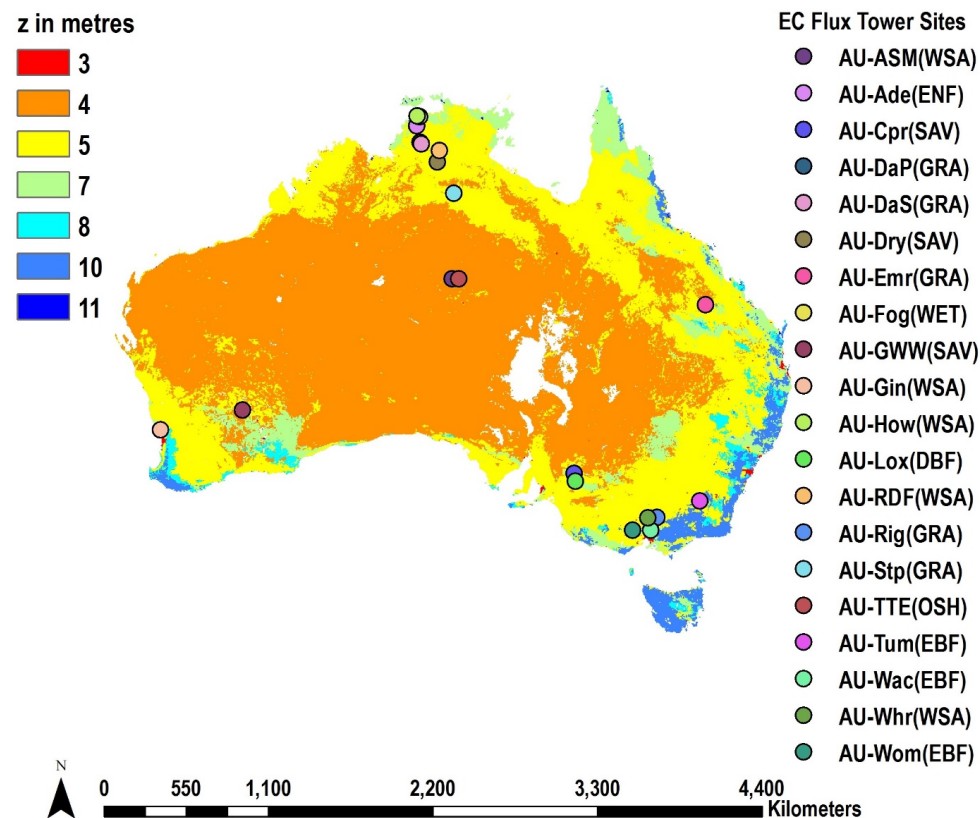

*Figure 2: Target height (z) in (m) above vegetation with location of Eddy Covariance flux towers and the land cover types;*

### 2.2.1 Hutson and Cass function with the two-point method

To determine the Hutson and Cass coefficients "a" and "b" (Eq. 14) for each 0.05° grid cell across Australia, we solve the pedotransfer with the two-point method. The two values used are the volumetric soil moisture ($\theta 1$ and $\theta 2$) at the field capacity and the wilting point soil water potentials ($\Psi 1$ and $\Psi 2$) of -10 kPa and 1500 kPa respectively. Combining both equations, we obtain the model parameters "a" and "b" for each 0.05° grid cell.

$$\Psi = a(\frac{\theta}{\theta_p})^{-b} \qquad (8)$$

$$\theta_p = 1 - (\rho / \rho_s) \qquad (9)$$

where $\Psi$ is the soil water potential (kPa); a (kPa) and b (-) are curve-fitting parameters; $\theta_p$ (-) is the porosity; $\rho$ (kg/dm$^3$) is the bulk density of soil; and $\rho_s$ =2.65 (kg/dm$^3$) is the mineral density.





### 2.2.2 Soil moisture

The soil moisture data used in this study are obtained from the European Space Agency's Climate Change
Initiative Soil Moisture Project (ESA CCI SM) at 0.25º and daily resolution available from 1978 to 2018 (Dorigo
et al., 2017), hereafter referred to as the ESA CCI SM. The ESA CCI SM consists of three products; Active,
Passive and Combined (Liu et al., 2012;Gruber et al., 2017). The ESA CCI SM is preferred in this study as it
offers the most suitable spatio-temporal resolution compared to other available soil moisture products. The
combined product is selected in this study as its algorithm unifies the Active and Passive products to have better
spatial coverage than either the Passive or Active products with more stringent quality control. While the
combined product has good spatial-temporal resolution for remote sensing applications, missing data are filled
through an average of the day before and after. Multiple-days data gaps are filled using multiple-days average.
The ESA CCI SM is also resampled at 0.05º resolution to be consistent with the spatial resolutions of the other
input data.

### 2.3 Transpiration

The MEP method requires specific humidity and temperature very close to the target surface. However due to the
difficulty of obtaining leaf surface temperature and specific humidity at regional scales, air temperature and air
specific humidity are used as surrogates. Air temperature and relative humidity data above canopy are obtained
from the interpolated field observations over Australia (Jeffrey et al., 2001). The Clausius-Clapeyron equation is
used in obtaining the specific humidity from air temperature and relative humidity.

### 2.4 Model Evaluation

For the evaluation of the MEP model results over Australia, data from 20 eddy covariance (EC) flux towers across
different land covers are used. The model performance is evaluated using six statistical metrics: the root mean
square error ($RMSE$), mean difference ($MD$), mean absolute error ($MAE$), Pearson's correlation coefficient (R),
Nash-Sutcliffe Efficiency ($NSE$) and Percent Bias ($PBIAS$),
$$RMSE = \sqrt{\frac{\sum_{n=1}^{N}(Q_n - \widehat{Q_n})^2}{N}} \qquad (10)$$

$$MD = \frac{\sum_{n=1}^{N}(Q_n - \widehat{Q_n})}{N} \qquad (11)$$

$$MAE = \frac{\sum_{n=1}^{N}|Q_n - \widehat{Q_n}|}{N} \qquad (12)$$

$$R = \frac{(\sum_{n=1}^{N}(Q_n - \overline{Q})(\widehat{Q_n} - \widehat{Q_n}))}{\sqrt{\sum_{n=1}^{N}(Q_n - \overline{Q})^2}\sqrt{\sum_{n=1}^{N}(\widehat{Q_n} - \widehat{Q_n})^2}} \qquad (13)$$

$$NSE = 1 - \frac{\sum_{n=1}^{N}(Q_n - \widehat{Q_n})^2}{\sum_{n=1}^{N}(Q_n - \overline{Q})^2} \qquad (14)$$



$PBIAS = 100 \times \frac{\sum_{n=1}^{N}(Q_n - \widehat{Q_n})}{\sum_{n=1}^{N} Q_n}$                    (15)

where $x_n$ and $y_n$ are observed and simulated daily ET values (mm); $N$ is the number of observed or simulated ET
values; $Q_n$ (mm) is the measured ET at day $n$; $\widehat{Q_n}$ (mm) is the simulated ET at day $n$; $\widetilde{Q_n}$ (mm) is the mean
simulated discharge at day $n$; and $\overline{Q}$ (mm) is the mean ET.

The MEP ET product at 5 km² resolution is validated across the 20 EC tower flux data with footprints ranging
from 100 m² up to about 2 km² depending on the measuring height of the EC system and vegetation height. The
effects of the differences in footprints of the EC towers and the data to be validated are not considered in this
study.
A three-product comparison (MEP, AWRA-L and MOD16) with the field data from the 20 EC flux towers across
Australia was conducted as part of this study. While the MEP and the AWRA-L models are produced on daily
timescales, the MOD16's highest temporary resolution is an 8-day product. For a direct comparison, MEP and
AWRA-L are aggregated to 8-day resolution. Since the MOD16 dataset has missing data points due to cloud cover
or sensor failures, the days with missing data are removed across all models and the EC tower data before
comparison.
Mean annual maps are produced for the three products between 2003 and 2013 with the MOD16 resampled to the
5 km² resolution to match that of the MEP and AWRA-L data for direct comparison for 280,000 pixels covering
the entire Australian using the R, RMSE, MAE and NSE statistical metrics.

**3.    Results and discussion**
**3.1 Mean spatial-temporal MEP ET Analysis**

The daily MEP evaporation and transpiration over Australia for 2003 – 2013 are relatively high in the northern
vegetated parts of Australia (Fig. 3a-b) and around the eastern coastline (Fig. 3b). Evaporation and transpiration
account for 38% and 62% of total ET, respectively, over Australia. ET is highest in the high rainfall shrub-lands
and forested regions in the northern Australia as well as around the coastline (Fig 3c). The west central parts of
Australia have the lowest ET with mean annual ET 440 mm for Australia for 2003-2013, while the mean ET along
the coastline exceeds 1000 mm for the same period.

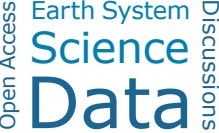

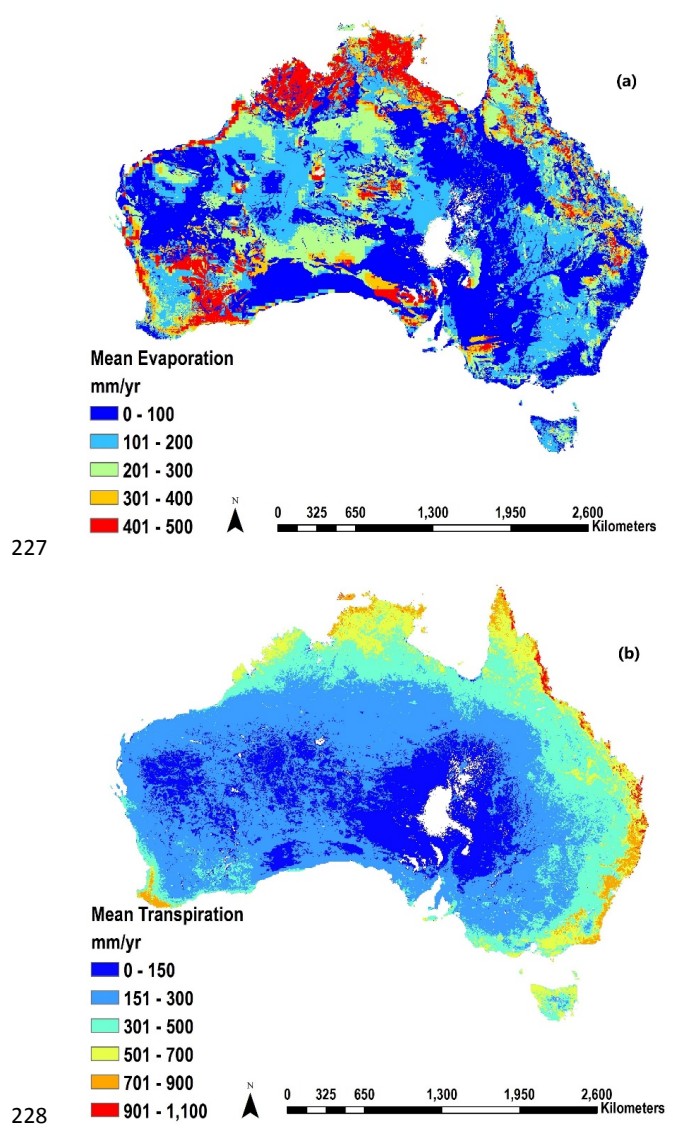



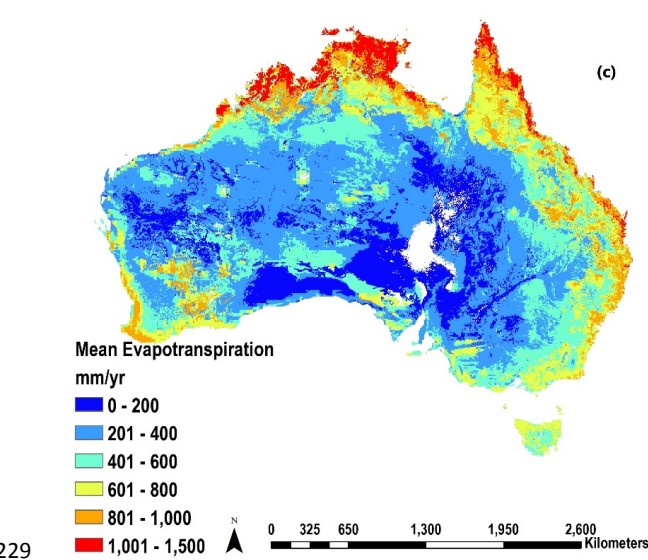


*Figure 3: (a) Mean evaporation; (b) Mean transpiration; and (c) Mean evapotranspiration in mm/yr for 2003-2013*

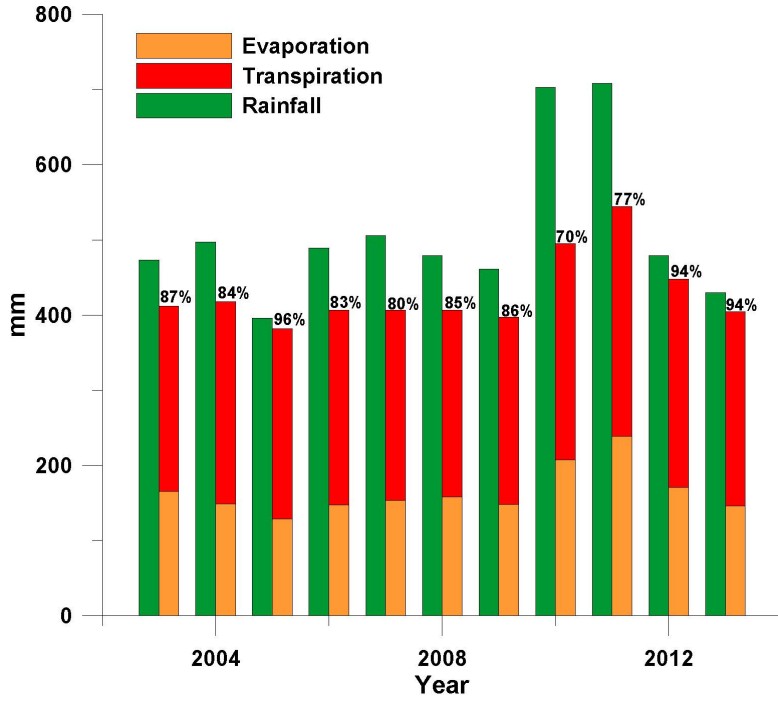


*Figure 3: MEP E &T vs Rainfall*



234 Annual ET fluctuates during the study period (Fig 3) with the correlations between annual evaporation and

235 transpiration and annual rainfall 0.94 and 0.84, respectively. Although the MEP model does not use rainfall as an

236 input, the strong correlation between rainfall and ET, the largest components of the hydrologic system in Australia,

237 suggests the MEP model captures the Australian hydrological system effectively. These results are consistent with

238 the findings of Jung et al. (2010) who observed a drop in the global evapotranspiration due to reduced ET over

239 Australia between 1998 and 2008. The reduction in ET over Australia can be seen through the "millennium

240 drought" years with the immediate increase in ET observed in 2010 at the end of the prolonged drought.








*Table 2: EC validation of the MEP, MOD16 and AWRA-L products. Eddy Covariance Tower Site name (Site Name); Fluxnet site ID and IGBP land*
*cover type (Site ID); Average observed ET at flux tower (OBS_ET); Root Mean Square Error (RMSE); Mean Absolute Error (MAE); Correlation*
*Coefficient (R); Percent Bias (PBIAS); EC sites citations*

| Site Name | Site ID | Obs_ET (mm/ 8 days) | RMSE (mm/ 8 days) | | | MAE (mm/ 8 days) | | | R | | | PBIAS (%) | | | Citations |
|---|---|---|---|---|---|---|---|---|---|---|---|---|---|---|---|
| | | | MEP | MOD16 | AWRA-L | MEP | MOD16 | AWRA-L | MEP | MOD16 | AWRA-L | MEP | MOD16 | AWRA-L | |
| Adelaide River | AU-Ade (WSA) | 15.34 | **9.06** | 11.09 | 9.65 | 7.04 | 9.22 | **5.73** | 0.64 | 0.57 | **0.71** | 26.18 | -34.38 | **22.26** | (Beringer, 2014g) |
| Alice Springs | AU-ASM (ENF) | 8.45 | **6.12** | 8.82 | 7.13 | **4.80** | 6.05 | 6.03 | **0.74** | 0.69 | 0.63 | **-6.78** | -62.1 | -23.9 | (Derek and James, 2014a) |
| Calperum | AU-Cpr (SAV) | 8.39 | **3.38** | 4.69 | 3.55 | **1.01** | 2.79 | 1.27 | 0.62 | 0.33 | **0.72** | **-12.04** | -33.25 | -15.15 | (Koerber, 2014) |
| Daly River Cleared | AU-DaS (GRA) | 18.6 | **4.62** | 9.74 | 6.05 | **3.63** | 8.21 | 4.43 | **0.88** | 0.74 | 0.78 | -12.23 | -38.6 | **0.21** | (Beringer, 2014f) |
| Daly River Savanna | AU-DaP (GRA) | 12.24 | 10.43 | 10.75 | **9.78** | 8.64 | 6.93 | **6.89** | 0.63 | 0.74 | **0.77** | 17.32 | **13.86** | 41.49 | (Beringer, 2014f) |
| Dry River | AU-Dry (SAV) | 19.55 | **9.95** | 13.63 | 12.58 | **4.7** | 8.14 | 5.02 | **0.62** | 0.43 | 0.58 | **-24.2** | -41.77 | -25.8 | (Beringer, 2014e) |
| Emerald | AU-Emr (GRA) | 11.56 | **5.69** | 5.96 | 9.91 | **4.22** | 4.35 | 7.32 | 0.47 | **0.48** | 0.43 | **-10.92** | -14.38 | 21.25 | (Schroder, 2014) |
| Fogg Dam | AU-Fog (WET) | 35.35 | **15.45** | 22.53 | 18.9 | **13.97** | 20.72 | 16.33 | 0.26 | 0.6 | **0.61** | **-35.71** | -58.4 | -42.79 | (Beringer, 2013b) |
| Gingin | AU-Gin (WSA) | 15.47 | 6.27 | 7.21 | **5.49** | 5.20 | 6.09 | **4.1** | 0.39 | 0.37 | **0.51** | **-3.0** | -36.49 | -17.02 | (Silberstein, 2015) |
| Great Western Woodlands, | AU-GWW (SAV) | 7.65 | **2.78** | 5.15 | 3.47 | **2.04** | 3.9 | 2.62 | **0.63** | 0.08 | 0.37 | 11.08 | -47.45 | **-11.06** | (Craig, 2014;Beringer, 2014d) |
| Howard Springs | AU-How (WSA) | 24.96 | **7.13** | 9.92 | 7.96 | **5.53** | 8.13 | 6.18 | 0.67 | 0.79 | 0.79 | **-3.2** | -30.0 | -9.87 | (Beringer, 2014c) |
| Loxton | AU-Lox (DBF) | 27.3 | 27.31 | **27.09** | 32.63 | 17.78 | **17.51** | 22.8 | **0.51** | 0.37 | -0.12 | -63.48 | **-60.0** | -82.7 | (Ewenz, 2008) |
| Red Dirt Melon Farm | AU-RDF (WSA) | 14.66 | **9.56** | 11.36 | 12.17 | **8.25** | 8.65 | 8.88 | **0.66** | 0.55 | 0.58 | **3.45** | -25.39 | 12.53 | (Beringer, 2013a) |
| Riggs Creek | AU-Rig (GRA) | 13.22 | 5.72 | 9.07 | **4.53** | 4.67 | 4.23 | **3.28** | 0.71 | 0.70 | **0.83** | -14.96 | -22.21 | **11.62** | (Beringer, 2014b) |
| Sturt Plains | AU-Stp (GRA) | 10.24 | **7.95** | 8.20 | 8.5 | 6.17 | 5.64 | **4.79** | 0.73 | **0.79** | 0.78 | 25.77 | -40.4 | **17.9** | (Schroder, 2014) |





| | | | | | | | | | | | | | | |
|---|---|---|---|---|---|---|---|---|---|---|---|---|---|---|
| Ti Tree East | AU-TTE (OSH) | 2.81 | 4.45 | **4.32** | 6.99 | 3.69 | **2.63** | 4.95 | **0.43** | 0.08 | 0.20 | 96.17 | **-42.34** | 146.08 | (Derek and James, 2014b) |
| Tumbarumba | AU-Tum (EBF) | 20.86 | 6.72 | 6.54 | **5.97** | 4.75 | 4.98 | **4.31** | 0.83 | 0.86 | 0.86 | -13.82 | 14.07 | **-6.57** | (Woodgate, 2014) |
| Wallaby Creek | AU-Wac (EBF) | 15.35 | 6.76 | 11.13 | **5.76** | 5.82 | 9.31 | **4.84** | **0.85** | 0.77 | 0.78 | 34.67 | 57.75 | **25.57** | (Beringer, 2014a) |
| Whroo | AU-Whr (WSA) | 13.73 | 6.51 | **5.08** | 5.86 | 5.09 | **4.10** | 4.52 | 0.54 | **0.59** | 0.46 | **-2.54** | -23.07 | -10.8 | (Beringer, 2014d) |
| Wombat | AU-Wom (EBF) | 23.28 | 8.24 | **5.13** | 7.45 | 7.11 | **4.16** | 6.02 | **0.89** | 0.88 | 0.81 | -30.12 | **-0.29** | -21.24 | (Beringer, 2014h) |




**3.2 MEP, MOD16 and AWRA-L performances at the Eddy Covariance flux sites**

The 20 eddy covariance flux tower sites used for the validation of the MEP, MOD16 and AWRA-L products include 8 land cover types according to the International Geosphere-Biosphere Programme (IGBP), i.e. 4-Evergreen Broadleaved Forest (EBF), 4-Woodland Savanna (WSA),4-Savanna (SAV), 1-Wetland (WET), 4-Grassland (GRA), 1-Evergreen Needle Forest (ENF), 1-Deciduous Broadleaved Forest (DBF), and 1-Open Shrubland (OSH). The MEP model outperforms the MOD16 at 15, 13, 14 and 16 sites measured by the RMSE, MAE, R and PBIAS metrics respectively. The MEP also performed better than the AWRA-L at 13, 11, 11 and 12 sites measured by the RMSE, MAE, R and PBIAS metrics, respectively. The MEP model also outperforms the MOD16 and AWRA-L measured by the average RMSE, MAE and R across the 20 EC flux sites. The average RMSE across the 20 EC flux sites for the MEP, MOD16 and AWRA-L are respectively 8.21, 9.87 and 9.22. The average MAE are respectively 6.21, 7.29 and 6.52 for the MEP, MOD16 and AWRA-L. The average correlations are 0.64, 0.57 and 0.61 for the MEP, MOD16 and AWRA-L, respectively. The MEP PBIAS was within 20% of the observed flux at 12 sites while the MOD16 and AWRA-L were within 20% of the observed flux at 4 and 10 sites, respectively.

Some consistency is seen across the models at many sites, with the three models seeming to perform best for the evergreen broadleaved forests with correlations ranging from 0.77 to 0.89 at the three sites. Similar correlation consistency of the models is obtained across the five grassland sites. Generally, the MOD16 underestimated ET significantly at most sites with 12 sites over 30%. Consistent underestimation is also observed across the Fogg Dam wetland site with the three models underestimating ET by 35% or higher. The MEP ET exhibited the lowest correlation at the Fogg Dam site. The Fogg Dam is a seasonally flooded wetland where water evaporation is a principal component of ET. However, due to the coarse resolution of the soil moisture data, the MEP model may not effectively capture the local evaporation. Less accurate ET estimates were also observed at the Loxton site by the three models with underestimation at least 60%. The flux data at the Loxton site appear unrealistic presumably caused by sensor failures suggested by 1800 mm ET while only 500 mm rainfall is recorded at the site.






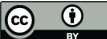






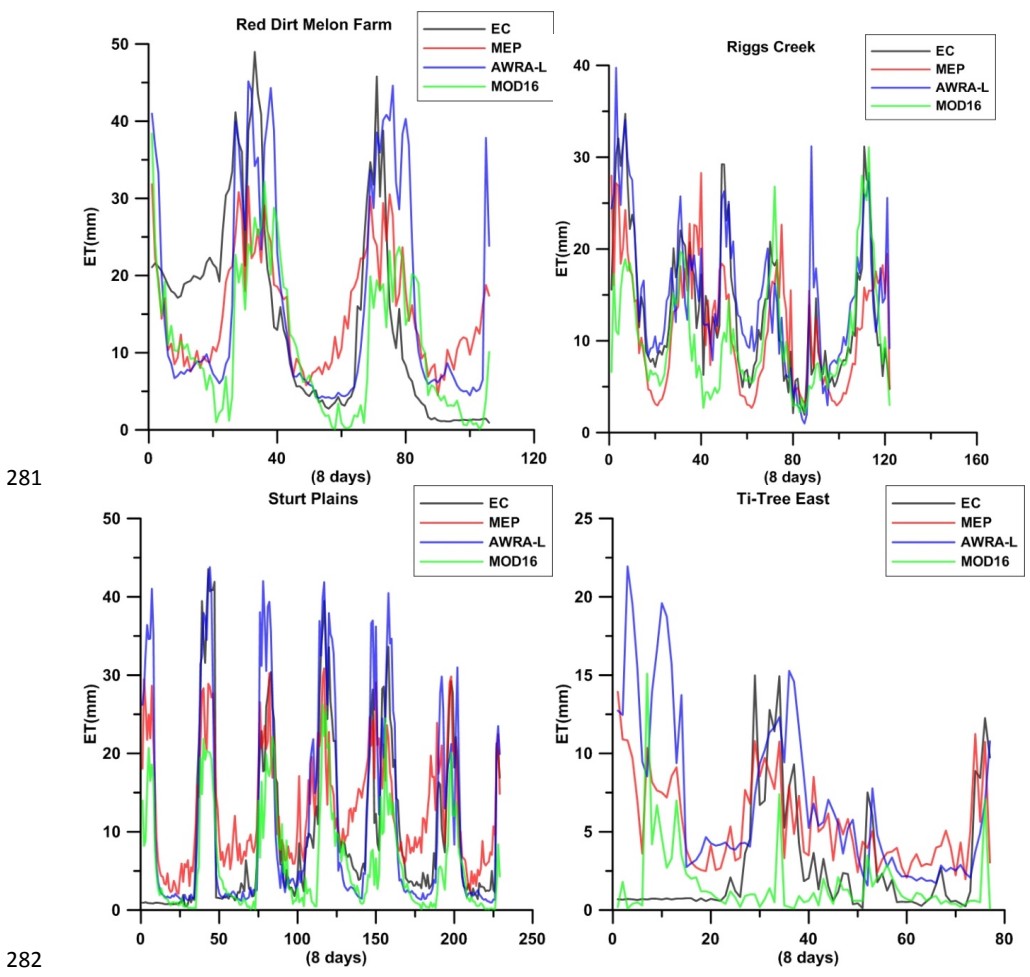



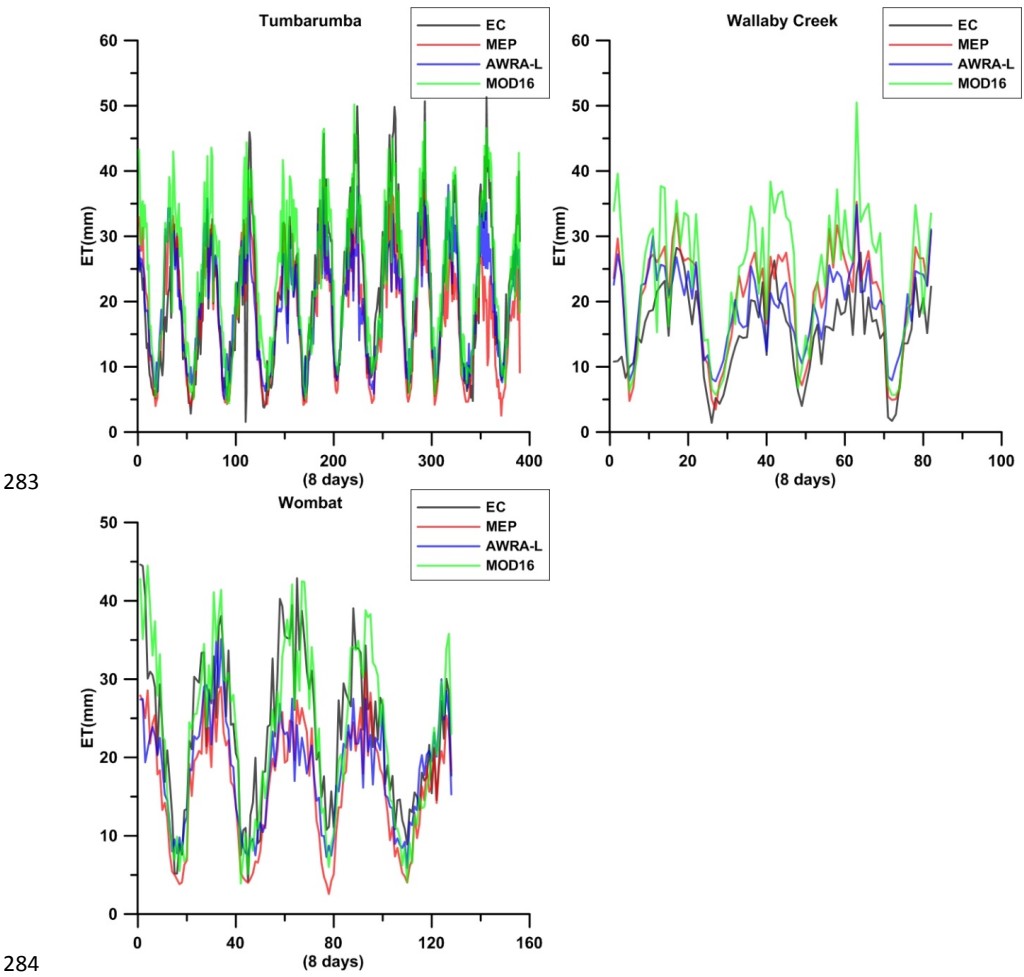



*Figure 4: Continuous plot of the MEP, EC, AWRA-L and MOD16 ET*

Fig. 4 shows that the MEP model reasonably captures the temporal trends of ET relative to the EC flux at most
sites. The MEP model appears to underestimate ET in the winter months and overestimate ET in the summer
months at the Whroo site. A possible reason for this trend in the MEP model is the wrong classification of the
vegetation at the Whroo site. The Whroo site, a box woodland revegetation from the gold mining era currently
covered with pasture and eucalyptus species vegetation, is incorrectly classified by the IGBP as an evergreen
broadleaved forest. The FPAR product used in partitioning net radiation between soil and canopy show large inter-
annual variation, leading to seasonal under- or overestimation of ET.





The MOD16 performs the best at forested sites showing consistent temporal patterns relative to the EC
observations. The calibrated AWRA-L model also effectively replicates the temporal trends across most sites and
outperforms the MOD16 at most sites.
The accuracy of the modelled ET is strongly affected by the estimated soil water potential using the pedotransfer
function. The difference in the footprints of the flux towers may also contribute to the underestimation of ET
particularly at flux tower sites with mixed vegetation.

**3.3 Comparison of the MEP, MOD16 and AWRA-L at Continental scale**

A continental scale comparison of the MEP, MOD16 and AWRA_L ET products was carried out after calculating
a mean annual ET over the study period from each product over the entire Australia. All 260,000 pixels of 5 km
resolution across the three models are used in the analysis. Annual mean ET over Australia from the MEP, MOD16
and AWRA_L products over the 11-year study period were calculated as 440, 262 and 428 mm, respectively. All
the corresponding cells were also used to calculate the correlation R, RMSE, NSE and MAE (Table 3). The spatial
agreements across the products was evident with all three products showing higher ET around the coastline and
lower ET in inland Australia. The NSE between the MEP and AWRA-L shows a better agreement than between
the MEP and MOD16 products, which have a negative NSE. The MAE and RMSE were also significantly lower
between the MEP and AWRA-L. The total ET from the MEP and AWRA-L appears more reasonable relative to
the annual rainfall over Australia (Fig 2). The annual MEP ET as a percentage of rainfall (Fig 2) is consistent with
other studies that about 90% of annual rainfall in Australia is returned to the atmosphere through ET (Chiew et
al., 2002;Prosser, 2011). Moreover, significant underestimation of ET by the MOD16 model was observed across
the flux tower sites.
Spatial analysis of the three products were also carried out using the percentage difference for MEP vs MOD16,
MEP vs AWRA-L and AWRA-L vs MOD16 (Fig 4). MEP ET was significantly higher than MOD16 ET for large
swaths of inland Australia while MOD16 was higher around the coastlines, particularly the eastern coastlines and
Tasmania. The underestimation of the MOD16 ET at the EC flux tower sites (section 3.3 showing that MOD16
underestimating ET at 17 of the 20 flux sites and by more than 30% in 12 of the sites) is confirmed as shown in
Fig. 4(a) and (c). The MOD16 performed better at the evergreen broadleaved forest tower sites close to the
coastline where it has better agreement with the MEP. However due to mixed performance of the MEP and



MOD16 model at the flux towers around the south-eastern coastline, it is difficult to draw a definite conclusion
on which model performs better. The percentage difference between the MEP and AWRA-L model has a narrower
range over large areas of Australia with both models within 50% for Australia. There are two large areas in the
south-central to Western Australia where the AWRA-L model significantly underestimates ET. The AWRA-L
ET is in the range of 1 – 10 mm/yr over large portion of Western Australia with numerous pixels having mean ET
less than 1 mm/yr between 2003 and 2013, which may be due to water balance errors in the AWRA-L algorithm.
The historic average precipitation in the partially vegetated region is in the range 200-500 mm/yr and it appears
implausible for ET to be less than 10 mm/yr. The large swath is also conspicuous in the AWRA-L and MOD16
percentage difference map (Fig 4c). The MOD16 model also produces higher ET than the MEP and AWRA-L
specifically in regions classified as evergreen broadleaved forests along the coastlines. The overestimation of
MOD16 at evergreen broadleaved forests has been documented in literature (Ruhoff et al., 2013;Hu et al., 2015).

*Table 2: The correlation coefficient (R), Root Mean Square Error (RMSE), Nash-Sutcliffe Efficiency (NSE) and Mean Absolute*
*Error (MAE) for comparison of the MEP, MOD16 and AWRA_L products over the entire Australia*

| | | RMSE (mm/yr) | | | | | MAE (mm/yr) | | |
|---|---|---|---|---|---|---|---|---|---|
| | | MEP | MOD16 | AWRA-L | | | MEP | MOD16 | AWRA-L |
| R | MEP | | 242 | 162 | NSE | MEP | | 203 | 126 |
| | MOD16 | 0.75 | | 205 | | MOD16 | -0.05 | | 187 |
| | AWRA-L | 0.77 | 0.86 | | | AWRA-L | 0.51 | 0.25 | |



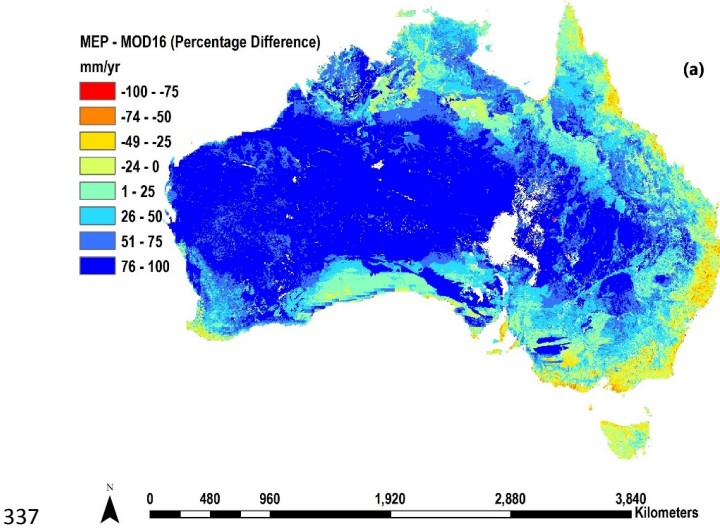



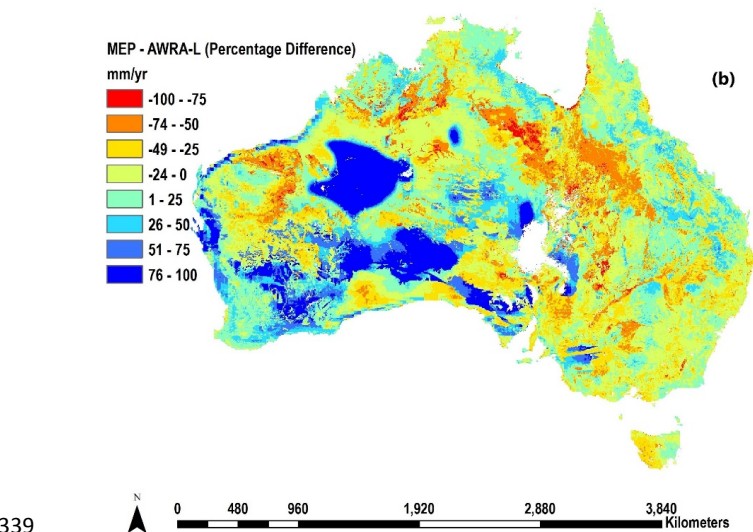






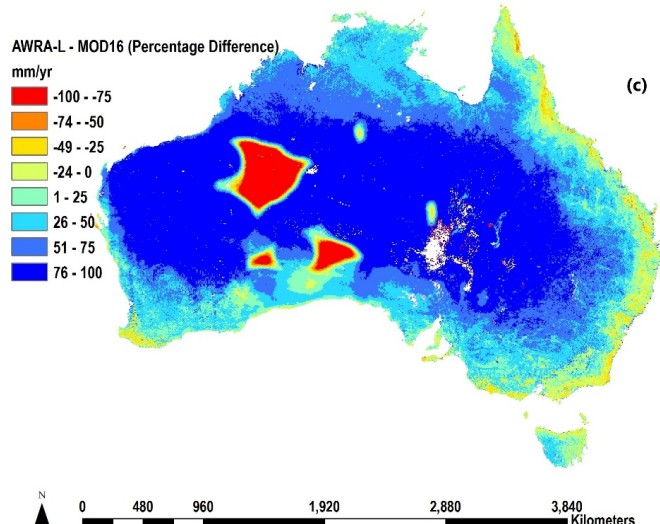


*Figure 5: Mean annual percentage difference between (a) MEP – MOD16; (b) MEP-AWRA-L; (c) AWRA-L-*


**3.4  Possible challenges with the MEP model**

The MEP model appears lacking spatial continuity, probably due to the use of pedotransfer functions to determine
the wilting point and field capacity, since surface specific humidity is a crucial input of the MEP model. Hence,
further improvement to the MEP model may be achieved by improving the parameterization of the pedotransfer
functions for each soil type.
Another challenge is the spatial resolution of soil moisture data for the regions where soil moisture is spatially
more variable. The low correlation of the MEP model in the Fogg Dam wetlands may be related to high spatial
variability of the soil moisture with intermittent flooding occurring at the site.









**4   Data Availability**

The produced daily evaporation and transpiration datasets over the entire Australia at the 5 km resolution for the
time period 2003 – 2013 (Abiodun et al., 2019) are publicly available at
http://dx.doi.org/10.25901/5ce795d313db8 or  from the direct download portal;
https://dap.tern.org.au/thredds/catalog/MEP/catalog.html
**5   Conclusion**

We have implemented the MEP model for estimating ET on a continental scale using readily available remote
sensing datasets to produce daily evaporation and transpiration at 5 km$^2$ resolution dataset over the entire Australia.
The MEP modelled ET was validated at 20 EC flux tower sites and compared to the MOD16 and AWRA-L model
ET. The MEP model outperforms both models at most EC flux sites with the AWRA-L model performing the
next best. The MEP ET has the best average RMSE, MAE, R and PBIAS across all 20 EC flux sites. The MEP
annual mean ET over Australia corroborates previous studies on the ET trend over Australia indicated by close
correlation between MEP ET and rainfall during and after the "millennium drought" period.
The MEP model is the simplest of the three models in terms of model formula and input data. This study shows
that the MEP model as a two-source surface energy balance model effectively estimates ET on regional scales
using fewer input data to produce evaporation and transpiration separately.
The MEP method has the potential to be further improved for modelling ET. Further study will seek to improve
the resolution of the MEP ET product while focusing on the development of a daily global MEP product.
**Appendix A**

The MEP model of evaporation and transpiration was derived from the dissipation function in Equation (A1) in
(Wang and Bras (2011)
$$D(E, G, H) \equiv \frac{2E^2}{I_e} + \frac{2G^2}{I_s} + \frac{2H^2}{I_a} \qquad \text{(A1)}$$
where $I_e, I_s,$ and $I_a$ are the thermal inertia relative to latent heat, ground heat and sensible heat flux, respectively,

$$I_s = \left(2.1\rho^{\left[1.2-0.02\left(\frac{\rho}{\rho_w}\right)100\theta\right]}e^{\left[-0.007\left(\frac{100\theta\rho}{\rho_w}-20\right)^2\right]} + \rho^{\left[0.8+0.02\left(\frac{\rho}{\rho_w}\right)100\theta\right]}\right)^{0.5} \times \left(\frac{\left(\frac{20\theta}{\rho_w}\right)\rho^2}{0.01}\right) \qquad \text{(A2)}$$
$I_s$ is parameterized as a function of soil moisture and water density and bulk density (Ma and Xue, 1990;Cai et
al., 2007) where $\rho_w$ is density of water (kg/m$^3$); $\theta$ is the soil moisture content of the soil (m$^3$/m$^3$);



$I_o = C_o \rho_a C_p \sqrt{kz} \left( \frac{kgz}{\rho_a C_p T_r} \right)^{\frac{1}{6}}$        (A3)
$C_o$ is the empirical constant characterizing the atmospheric stability (Businger et al., 1971): $C_o = 1.7$ Unstable,
1.2 Stable; $\rho_a$ is the density of air (Kgm$^{-3}$); $k = 0.4$ the von Kármán constant; $z$ is the distance above the target
surface for which the Monin-Obukhov similarity theory is valid (m); $g = 9.8$ m/s$^2$ the acceleration due to gravity;
$T_r$ ($\sim$ 300 K) is an atmospheric reference temperature;
$I_a = I_o |H|^{-\frac{1}{6}}, I_e = \sigma I_a,$        (A4)
where $\sigma$ is defined in Equation 2
In the MEP equation over vegetated land surface in Wang and Bras (2011), the reciprocal Bowen ratio; $\beta(\sigma) =$
$6\left( \sqrt{1 + \frac{11}{36}\sigma} - 1 \right)$, was introduced to represent the target surface conditions as a function of specific humidity
and temperature. Hence, the MEP flux equations over vegetated land can be written as,
$E_v = \frac{R_{n\_v}}{1 + \beta(\sigma)_v^{-1}}, H_v = \frac{R_{n\_v}}{1 + \beta(\sigma)_v}$        (A5)
At regional scales where air specific humidity and air temperature are used as surrogates of canopy surface specific
humidity and temperature, $\beta(\sigma)$ in equation A5 is replaced with $\sigma$
$\theta @FC = 7.561 + 1.176Clay - 0.009843Clay^2 + 0.2132Silt$        (A6)
$\theta @PWP = -1.304 + 1.117Clay - 0.009309Clay^2$        (A7)
Pedotransfer functions in Equations A6 and A7 are used to determine the soil moisture content at field capacity
and permanent wilting point as the inputs into the Hutson and Cass model in Equation. FC is the field capacity (-
); Clay and Silt are the clay and silt fraction of the soil; and PWP is permanent wilting point (-).

**Acknowledgement**

We would like to acknowledge the invaluable advice of Dr John Hutson in the preparation of this manuscript.
This work used eddy covariance data acquired and shared by the FLUXNET community, including these
networks: OzFlux-TERN. The ERA-Interim reanalysis data are provided by ECMWF and processed by LSCE.
The FLUXNET eddy covariance data processing and harmonization was carried out by the European Fluxes





Database Cluster, AmeriFlux Management Project, and Fluxdata project of FLUXNET, with the support of
CDIAC and ICOS Ecosystem Thematic Center, and the OzFlux, ChinaFlux and AsiaFlux offices.

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
