# Peer review of "A Maximum Entropy Production Evaporation - # Transpiration Product for Australia"

_Earth System Science Data, 2019_

## Referee Comment (RC1) · Anonymous Referee #1 · 17 Sep 2019

This paper addresses an important but complex issue, namely to estimate the rates of evaporation from soils and transpiration from vegetation over large areas (here the whole of Australia). However, this contribution suffers from multiple problems. The following comments are intended to highlight those issues and help the Authors to revise their manuscript.

Major comments:

1 Each and every input data set should be systematically characterized in terms of its nature, units of measurements, spatial and temporal resolution, accuracy (error bar), and source. Much of that information is currently lacking for most of the inputs used.

2 Assuming that the heat flux into the ground $G$ is null for vegetated areas is a crude

approximation at best. At the very least, this should be supported by ample empirical evidence.

3 Soil moisture, derived from a mix of space-based instruments, is arguably the most important source of information in this study, and likely the driving factor that guarantees reasonable results. Yet, there is no discussion of the accuracy of that product, nor of the dependency of the output on that accuracy.

4 The Authors correctly point out, in the introduction, the difficulty of mixing and merging (let alone assessing) input data obtained at widely different spatial and temporal scales and resolution. Yet, little or no discussion of this key issue appears in the paper: Satellite-derived soil moisture data comes at a spatial resolution of 25 km. Re-sampling it at 5 km may be convenient programmatically, but that does not mean that this information is suddenly available (and reliable) at this finer resolution. Similarly, saying that field data obtained from flux towers have a footprint "ranging from 100 $m^2$ up to about 2 $km^2$ depending on the measuring height of the EC system and vegetation height" (lines 204–205) has to be taken at face value, because no evidence is provided to support such a claim. Hence, comparing those outcomes and claiming that one validates the other is a gross oversimplification of the matter.

5 The paper does not discuss the concept of potential evaporation (PE: the maximum rate of evaporation when water supply is not a limiting factor), nor does it provide a map of the annual mean precipitation over Australia. Yet, both of those variables constitute caps which evapotranspiration (ET) is not supposed to exceed... A map of precipitation for Australia is available from the Bureau of Meteorology at (`http://www.bom.gov.au/jsp/ncc/climate_averages/rainfall/index.jsp`). Comparing the ET output of MET with that map, it is not obvious that the mean evapotranspiration (first Figure 3 on p. 12) is everywhere lower than the precipitation at the same location...

6 The MEP model mobilizes multiple input sources as well as a large number of equations or algorithms with (often fixed) empirical parameters. The latter may have been derived for other locations or time periods. It is not clear what is achieved by this complexity, or to what extent each source actually contributes to the final outcome. Could a much simpler model account fo the bulk of the variability? What is gained by the complexity, especially when it involves constant parameters? When developing large models like the MEP, it is essential to document the relative importance of the main inputs and the sensitivity of the outputs to those inputs.

7 Although the study is carried out over a decade (2003–2013), the manuscript does not discuss very much the time evolution of ET during that period. Figure 4 does show some time series over periods ranging from 80 to about 350 days, but it is not clear whether those are for a particular year or averaged over the entire period (and if so, where do these occur in the calendar year?). In any case, why are those time series exhibited over such different periods? For similar reasons, the enigmatic second Figure 3 (on p. 12) barely addresses the issue of confronting the time evolution of ET generated by MEP with actual measurable evidence. Options include comparisons with agricultural output, drought and flood periods, or any other biogeophysical variable that could betray the impact of ET fluctuations during that decade.

8 Lastly the legends of the Tables and Figures are so minimalist as to be largely useless to understand their contents. Please provide essential information to understand and appreciate the nature and contribution of those displays.

Editorial and substantial remarks (in sequential order):

- Lines 56–63: Replace "accepted" by "used": MODIS products are widely used because they are available and accessible, and because standard tools exist to manipulate the large data sets. The word "accepted" implies a vetting of the quality and performance of the product which may or may not apply, as indicated by the Authors themselves (lines 58 to 64).

- Line 68: This model is called 'Maximum Entropy Production', but there is no indication

about what is maximized, or what connection may exist with the concept of entropy.

- Line 71: Clarify that MEP requires the specific humidity *of the air*.

- Lines 94 and 102: Indicate explicitly that $q_s$ refers to the air specific humidity: the expression 'surface specific humidity' can be ambiguous.

- Lines 94–95: The text mentions $q$ but the equation uses $q_s$. Adding to the confusion, the subscript $s$ is used throughout the paper to designate soil variables, while $q$ appears to be an atmospheric variable...

- Lines 95–97: Equations (3) and (4) show that the heat flux into the ground $G$ and the latent heat flux $E_s$ are both proportional to the sensible heat flux from the soil surface to the atmosphere $H_s$, but there is no indication about how the latter is actually estimated. Is it assumed to be the residual in Equation (1)? If so, what about the sensible heat flux from the plant canopy to the atmosphere $H_v$? This system of equation does not appear to be complete or energy conserving as stated. What is the accuracy of those equations, and to what extent are they (in particular the inverse Bowen ratio $\beta_s$) relevant and applicable to the Australian environment, given the presence of empirical coefficients, which may have been fine-tuned for US conditions? Where are the values of $q_s$ and $T_s$ coming from, what are their accuracy, spatial and temporal resolutions?

- Lines 111–114: Comments on the flowchart in Figure 1:

* This Figure describes how the various input data sets are processed by MEP to generate the desired output, evapotranspiration. Yet, none of those boxes appear to require any of the fluxes $H$, $E$ or $G$ mentioned earlier: only $R_n$ is discussed or used subsequently. So what is the relation, if any, between the materials and equations described between lines 89 and 108, on one side, and the rest of the paper?

* Figure 1 distinguishes between "soil surface relative humidity" and "soil surface specific humidity", yet the equations or algorithms to derive those variables are nowhere to be seen...

* According to Equation (2), the variable soil sigma ($\sigma_s$) is derived from both soil surface temperature $T_s$ and atmospheric specific humidity $q_s$, yet, the Figure does not show this latter dependency.

- Line 118: What are the source, accuracy, spatial and temporal resolutions of $R_n$?

- Line 119: FPAR is mentioned here (on page 6), but that product does not appear as an input in Figure 1. Also, it is not clear what its role is at this point (this is only clarified on page 20!–see the comment on Line 292 below).

- Line 121–125: The text appears to use the expressions "vegetation cover" and "vegetation fraction" as synonyms. This is confusing, as "vegetation cover" could be understood as "land cover", which typically refers to the type of ecosystem (forest, savanna, etc.) while "vegetation fraction" hints to the more appropriate concept of "fractional vegetation cover". This latter variable is itself generally poorly defined (and hard to estimate when the Leaf Area Index is less than 3), as is often the case in arid environments like Australia. In any case, if the fractional vegetation cover $F_c$ is derived from FPAR, then Figure 1 should show FPAR in the crossed-out box, and $F_c$ should appear in a separate box.

- Lines 59–61 and 131: There is an internal inconsistency in first decrying the poor quality of the MODIS temperature product and in using it nevertheless. This Reviewer cannot comment on the value of this product, but if the Authors estimate it is incorrect, they should use another one.

- Lines 131–134: The rationale for using the lowest $T_s$ during the month is dubious: what is the accuracy and reliability of that product, generated on an 8-day basis (according to Line 59), if the area of interest happens to be overcast on successive 8-day periods? And even if $T_s$ is always observed at least once a month, that measurement would necessarily occur on a clear (relatively hotter) day, so there is still a bias towards high temperatures during cloudy days. What is the possible impact of that bias?
- Lines 135–149: This whole paragraph appears very confusing, because it amounts to a somewhat disparate assemblage of algorithms and equations found in the literature, using multiple empirical coefficients which may or may not be applicable to Australia. Why mention methods that are not used? And again, those tools depend on additional soil properties (whose accuracy, spatial and temporal resolutions are unknown, by the way) apparently obtained from the Australian Soil Resource Information System (AS-RIS), though the latter is not reported as an input in Figure 1. As the Authors rewrite this paragraph, they would be well advised to describe the necessary steps in logical order, to explicitly provide the full equations, to indicate clearly what inputs are needed, where they come from, their spatial and temporal resolutions, accuracy, etc.

- Line 140: The text appears to indicate that the values of the soil water content at wilting point and at field capacity are fixed in space and time to -1.5MPa and -10kPa, respectively. What evidence is there that this is reasonable, given the high diversity of soil types and properties?

- Line 145: What exactly is implied here by "modest data requirement and relative accuracy"? Constant values would be even simpler... This is not an acceptable rationale: the selection of models and parameterizations should be guided by strict requirements in terms of accuracy and performance to achieve a particular objective, not in terms of data volume or approximate value. Besides, how small should a database be to be "moderate", and what is the benchmark to evaluate the "relative accuracy"?

- Line 151: The authors introduce the concept of "distance $z$ above the target surface for which the Monin-Obukhov similarity theory is valid in the formula of the thermal inertia of turbulent air above soil surface", but none of those Equations are provided. A Table of (fixed) values is provided, and a map of $z$ over Australia is produced, but what are the significance and implications of this variable in the MEP model? Since 90% or more of that continent is covered by low vegetation anyway, does it matter to consider this parameter?

[Figure]

- Line 155: What is the scientific basis for assigning those particular values in Table 1? Note also that the text refers to $z$ as a "target distance", while the legend appears to refer to a "target surface".

- Line 158: In Figure 2, why is there a blue block corresponding to $z$ values of 11 m when Table 1 does not show any land cover with that value? And why is the legend to this figure labeling $z$ as the 'target height (z)', which would be normally understood as the height of vegetation, rather than the theoretical concept mentioned above?

- Line 161: The text refers to Eq. 14, but that Equation (on line 197) has nothing to do with "the Hutson and Cass coefficients a and b": this probably refers to an equation in another paper...

- Line 166: Where does the soil density come from? What is its accuracy and spatial resolution?

- Lines 171–181: The ESA CCI Soil Moisture (SM) product is delivered at the spatial resolution of $0.25°$. Resampling it (by duplication or interpolation) on a grid at $0.05°$ may be programmatically convenient or necessary, but that does not change the intrinsic nature or properties of the product! Also, which version of that product is actually used here, since it has already been released 6 times?

- Lines 186–187: Please provide an explicit description of the interpolation procedure as well as the formula to calculate the specific humidity of the atmosphere using the Clausius-Clapeyron equation: any reader should be allowed to duplicate the work without having to guess which tools, techniques or equations should be used.

- Line 200: The text mentions the variables $x_n$ and $y_n$, but there are no such variables in the equations above, and it is not clear what distinguishes them anyway.

- Line 201: The diacritical marks on top of the $Q$ symbols are barely distinguishable on this line, and almost impossible to differentiate in the Equations above (Lines 193–198). Please use other signs or reformat or enlarge the Equations.

- Lines 204–205: The wording here may be misleading: The "MEP ET product" may be technically available "at 5 km$^2$ resolution", but to the extent the space-derived soil moisture data is the dominant input, the actual spatial resolution may be closer to 25 km by 25 km or 625 km$^2$, as noted in the comment above (Lines 171–181). Similarly, the statement "tower flux data with footprints ranging from 100 m$^2$ up to about 2 km$^2$" is purely gratuitous, in the absence of evidence, or algorithms, to support these estimates. Hence comparing these products and claiming that this is a validation is unwarranted, on the basis of the information presented here.

- Lines 229–233: Please note that there are two Figures labeled "Figure 3" on this page.

- Line 231: In the second Figure 3, what is the source of those data? What is the area concerned (the whole of Australia)? Is this the result of an accumulation of data from rain-gauges, or a satellite product, or a reanalysis? What is the accuracy of those precipitation estimates? What is the spatial representativity of the precipitation data? Do they provide a spatial coverage comparable with the gridded data of MEP? And of course, most importantly, what is the error bar associated with the MEP estimates?

- Line 247: The legend of Table 2, or the text (or both), should indicate that the numbers set in bold face in this table point to the best performing method to evaluate the evapotranspiration rate at the various Eddy Covariance (EC) sites, according to the criteria indicated in the table header.

- Line 292: At long last, the text reveals that "The FPAR product [is] used in partitioning net radiation between soil and canopy". However, this is basically incorrect and inappropriate: FPAR is a measure of the effectiveness of the vegetation in absorbing photosynthetically active radiation (PAR), not in any way an indicator of the fractional vegetation cover. A particular value of FPAR derived from satellite observations could be obtained from a wide range of ecosystems with widely varying $F_c$ and Leaf Area Index (LAI), not to mention other possible factors.
- Lines 307 and 335: Table 3 is mentioned in the text before Table 2, and no Table is labeled 3 in this document. However, there is a Table 2 starting on Line 244, and another one starting on Line 335...

- Line 312: The Figure referred to by "as a percentage of rainfall (Fig 2)" exhibits the spatial distribution of $z$, not precipitations. In fact, there is no precipitation map in the paper, although one would be useful, as noted above.

- Line 346: The statement "The MEP model appears lacking spatial continuity, probably due to the use of pedotransfer functions..." is invalid. Those functions are just mathematical formulae, fitting functions: they cannot by themselves generate spatial discontinuities. If the MEP model outputs appear spatially discontinuous, it must be because the soil moisture input data themselves are discontinuous, or because of model (coding) errors. For this reason too, it would be useful to conduct sensitivity analyses to establish to what extent each input and algorithmic parameter or equation contributes to the outcome. If the discontinuity does arise from the intrinsic variability of the soil moisture data, no amount of tuning of the pedotransfer functions will reduce those discontinuities. A contrario, if the soil moisture data are reasonably homogeneous to start with, then there may indeed be a problem with the way those functions, or other aspects of the model, are implemented. In either case, the conclusion that "Hence, further improvement to the MEP model may be achieved by improving the parameterization of the pedotransfer functions for each soil type." is currently unfounded.

- Line 351: The text states "The low correlation of the MEP model" but does not indicate with what MEP does not correlate well.

- Lines 410–411: The acknowledgment mentions the use of FLUXNET and ERA-Interim reanalysis data: Why are those data sets not mentioned in the text and appear in Figure 1?

---

## Referee Comment (RC2) · Anonymous Referee #2 · 23 Nov 2019

I completed the review of the manuscript titled "A Maximum Entropy Production Evaporation - Transpiration Product for Australia". Here, the authors used the maximum entropy production (MEP) for evaporation and transpiration estimation at 5 km resolution for entire Australia. They compared the ET estimation between the eddy covariance tower, MEP method, MOD16 and Australian Water Resource Assessment Landscape (AWRA-L). I think the authors produced useful data for Earth Science studies. However, this manuscript is not ready for publication and the mechanics and discussions of the manuscript need improvement. They are given below:

The results and discussion needs lots of work. It will be good to describe the spatiotemporal dynamics of MEP E and T in detail for 2003-2013 period and connect them with local and regional hydroclimatic fluctuations. Possible challenges can be expanded

using uncertainty of the input dataset. What can we learn from these MEP products? What is the new knowledge this dataset can inform us that we do not know right now? What types of future studies this MEP dataset can generate for our scientific community?

Figure 3 caption is inadequate (Mean maps of which product??? MEP?). Presenting the mean annual ET is not enough to claim the conclusion. I think maps of standard deviation, ET map of max ET (the year of Maximum at study area scale) and min ET map (the year of Minimum ET at study area scale) are required.

Figure numbering got messed up. Figure 3 was labeled twice. As a result, it was difficult to follow the manuscript.

Line 111: What is the source of vegetation fraction (Not provided/explained?)?

Figure 1: Did you use rainfall, humidity, and temperature from one station? If multiple stations, then how did you distribute/interpolate across the entire content? How much is the spatial variability of precipitation across Australia?

Please add a discussion regarding the propagation of the uncertainty or errors in the input dataset to your MEP product. I believe the vegetative fraction and soil moisture have quite a bit of uncertainty.

What is the extent of footprint for eddy covariance tower estimates? How this extent compare to the pixel size of the MEP product? Please add this discussion somewhere.

Line 236-238: I don't see any ET reduction between 2003 and 2008 in figure 3. How this is consistent with Jung et al., (2010) Line 309: NSE is not reported in this manuscript. Table 2 does not show any NSE.

---

## Author Comment (AC1) · 1 May 2020

The authors are immensely grateful for the invaluable review, corrections and suggestions given by the reviewer.

| Reviewer comment | Author response |
|---|---|
| | Signficant concerns of the reviewer appears to arise from non-familiarity with the postulation, development and derivation of the MEP theory which have been comprehensively published by (Wang and Bras, 2009;Wang et al., 2010;Wang and Bras, 2011;Wang et al., 2014). This is understandable as the MEP is a relatively new method of ET estimation. The focus of this manuscript is not to re-derive the MEP method, it is to apply and create a continental scale product based on a method that has been established, tested and validated. |
| Each and every input data set should be systematically characterized in terms of its nature, units of measurements, spatial and temporal resolution, accuracy (error bar), and source. Much of that information is currently lacking for most of the inputs used | The authors will include a table which details the input datasets nature, units, resolutions and the citation of sources. |
| Assuming that the heat flux into the ground G is null for vegetated areas is a crude approximation at best. At the very least, this should be supported by ample empirical evidence | The authors did not assume Ground heat flux (G) is null in vegetated areas. The MEP Evaporation equations in Eqn 2-4 over soil calculate for G while the MEP Eqns 5 and 6 for Transpiration neglects G on the canopy as it will have been accounted for under the canopy through the evaporation equations. See (Wang and Bras, 2011).

 The manuscript will be updated to ensure clarity of the above point. |
| Soil moisture, derived from a mix of space-based instruments, is arguably the most important source of information in this study, and likely the driving factor that guarantees reasonable results. Yet, there is no discussion of the accuracy of that product, nor of the dependency of the output on that accuracy. | The soil moisture is an important variable used in the MEP method in this study. However, it is only used in the Evaporation calculations. The accuracy of this product (ESA CCI Soil moisture v 04.4) (Dorigo et al., 2017) has not been extensively discussed as there are manuscripts dedicated to the validation and review of this soil moisture products (Dorigo et al., 2015;McNally et al., 2016;An et al., 2016;Dorigo et al., 2017). However, a discussion regarding this will be included in the updated manuscript. |
| The Authors correctly point out, in the introduction, the difficulty of mixing and merging (let alone assessing) input data obtained at widely different spatial and temporal scales and resolution. Yet, little or no discussion of this key issue appears in the paper: Satellite-derived soil moisture data comes at a spatial resolution of 25 km. Re-sampling it at 5 km may be convenient programmatically, but that does not mean | The challenges surrounding scales and resolutions will be discussed in the updated manuscript. The statement in lines 204 – 205 has been removed. However, It was mentioned in lines 205 and 206, that the effects of the footprint of the obtained EC data from FLUXNET is not considered in this study. The authors considered that similar studies globally like we have undertaken, have compared ET products to EC tower ET results without |

| | |
|---|---|
| that this information is suddenly available (and reliable) at this finer resolution. Similarly, saying that field data obtained from flux towers have a footprint "ranging from 100 m2 up to about 2 km2 depending on the measuring height of the EC system and vegetation height" (lines 204–205) has to be taken at face value, because no evidence is provided to support such a claim. Hence, comparing those outcomes and claiming that one validates the other is a gross oversimplification of the matter | footprint analysis (Jin et al., 2011;Velpuri et al., 2013;Mu et al., 2011a;Hu et al., 2015). This is due to the difficulty of multi-scaling analysis as well as availability of footprint data from the FLUXNET daily analysis. The use of EC flux tower data in regional/continental scale studies such as ours give an indication of the relative closeness or divergence to ground-based measurements in location with EC data thereby giving a certain degree of confidence to regional to continental scale products such as this. Otherwise there would be no way to give any indication of accuracy. |
| The paper does not discuss the concept of potential evaporation (PE: the maximum rate of evaporation when water supply is not a limiting factor), nor does it provide a map of the annual mean precipitation over Australia. Yet, both of those variables constitute caps which evapotranspiration (ET) is not supposed to exceed... A map of precipitation for Australia is available from the Bureau of Meteorology at (http://www.bom.gov.au/jsp/ncc/climate_averages/rainfall/index.jsp). Comparing the ET output of MET with that map, it is not obvious that the mean evapotranspiration (first Figure 3 on p. 12) is everywhere lower than the precipitation at the same location.. | One of the unique features of the MEP method is the fact that the method does not require potential Evaporation to compute ET (See (Wang and Bras, 2011). Also, regarding the use of the annual mean precipitation map suggested by the reviewer, this was initially considered by the authors, but discovered to be inappropriate as the map from the BOM was an average of the period between 1961 -1990. However, we have produced the ET product over Australia for the period of 2003 – 2013, which were especially dry due to the millennium drought. Hence the possible differences observed by the reviewer. Since the map was not representative of the study period, it was decided by the authors to be excluded from the manuscript for comparison. |
| The MEP model mobilizes multiple input sources as well as a large number of equations or algorithms with (often fixed) empirical parameters. The latter may have been derived for other locations or time periods. It is not clear what is achieved by this complexity, or to what extent each source actually contributes to the final outcome. Could a much simpler model account for the bulk of the variability? What is gained by the complexity, especially when it involves constant parameters? When developing large models like the MEP, it is essential to document the relative importance of the main inputs and the sensitivity of the outputs to those inputs | The goal of this paper was to create evaporation and transpiration products over the entire Australia using the MEP method. All empirical equations used in this manuscript were developed for Australian conditions. As with all regional to continental scale ET products (MOD16, LSA-SAF MSG, SSEBop ET), there are required data which are not available at regional scale e.g specific humidity, surface roughness, stomatal conductance etc. The authors attempt to obtain such parameters through the use of empirical equations or other derived algorithms. While it is acknowledged that these estimations will propagate some error through the product, it is often the most practical way. Hence the use of ground-based measurements to compare, to determine if the results are acceptable or not, which is what we have done in this study. The importance of the principal inputs and their sensitivity have been documented in the MEP development articles (Wang and Bras, 2009;Wang et al., 2010;Wang and Bras, 2011;Wang et al., 2014). |
| Although the study is carried out over a decade (2003–2013), the manuscript does not discuss very much the time evolution of ET during that period. Figure 4 does show some time series over periods ranging from 80 to | The evolution of ET during the study period will be included in the updated manuscript. |

| | |
|---|---|
| about 350 days, but it is not clear whether those are for a particular year or averaged over the entire period (and if so, where do these occur in the calendar year?). In any case, why are those time series exhibited over such different periods? For similar reasons, the enigmatic second Figure 3 (on p. 12) barely addresses the issue of confronting the time evolution of ET generated by MEP with actual measurable evidence. Options include comparisons with agricultural output, drought and flood periods, or any other biogeophysical variable that could betray the impact of ET fluctuations during that decade | The time series in figure 4 was a comparison between the EC tower data and the three products (MEP, AWRA-L and MOD16). The analysis was constrained to the days where the EC data intersected the three products between 2003 -2013, hence the range from 80 – 350 days.

Due to the comparison of three products with the EC tower sites, only days where there was data across the four datasets being compared was used. Hence it is impracticable to include dates.  In the Figure 3, we will include a comparison of the MEP yearly average with the yearly average of another continental scale ET product. |
| Lastly the legends of the Tables and Figures are so minimalist as to be largely useless to understand their contents. Please provide essential information to understand and appreciate the nature and contribution of those displays | We are sorry, we recognize we have been too brief in our figure captions. We will update these in the revised manuscript to be much more explanative. |
| Lines 56-63: Replace "accepted" by "used": MODIS products are widely used because they are available and accessible, and because standard tools exist to manipulate the large data sets. The word "accepted" implies a vetting of the quality and performance of the product which may or may not apply, as indicated by the Authors themselves (lines 58 to 64). | We agree, this will be updated in the manuscript. |
| - Line 68: This model is called 'Maximum Entropy Production', but there is no indication about what is maximized, or what connection may exist with the concept of entropy. | This model was named "Maximum Entropy Production method" by the authors of the method (Wang and Bras, 2009;Wang and Bras, 2011) and the derivation and conceptualization have been extensively documented. We have tried to make this a data paper as much as possible without repeating the whole conceptualization in this manuscript. |
| Line 71: Clarify that MEP requires the specific humidity of the air | Indeed, the specific humidity of air at the target surface (soil or canopy) is a requirement of the MEP method. This will be updated in the manuscript. |
| - Lines 94 and 102: Indicate explicitly that qs refers to the air specific humidity: the expression 'surface specific humidity' can be ambiguous | This refers to soil or canopy surface. Soil surface for the evaporation equation and canopy surface for the transpiration equation.

This will be clarified and updated in the manuscript. |
| - Lines 94–95: The text mentions q but the equation uses qs. Adding to the confusion, the subscript s is used throughout the paper to designate soil variables, while q appears to be an atmospheric variable... | $q_s$ is specific humidity of air at the target surface (soil or canopy)

This will be clarified and updated in the manuscript. |
| - Lines 95–97: Equations (3) and (4) show that the heat flux into the ground G and the latent heat flux Es are both proportional to the sensible heat flux from the soil surface to the atmosphere Hs, but there is no indication about how the latter is actually estimated. Is | The MEP method is completely derived mathematically through the lagrange multiplier method by optimising the dissipation function in (Wang and Bras, 2011). No parameters in the MEP have been fine tuned for any specific location. The authors avoided re-deriving the |

| | |
|---|---|
| it assumed to be the residual in Equation (1)? If so, what about the sensible heat flux from the plant canopy to the atmosphere Hv? This system of equation does not appear to be complete or energy conserving as stated. What is the accuracy of those equations, and to what extent are they (in particular the inverse Bowen ratio βs) relevant and applicable to the Australian environment, given the presence of empirical coefficients, which may have been fine-tuned for US conditions? Where are the values of qs and Ts coming from, what are their accuracy, spatial and temporal resolutions? | MEP equation in this manuscript as this has been comprehensively covered in (Wang and Bras, 2009;Wang and Bras, 2011;Wang et al., 2014).

The equations 2, 3 and 4 are solved using a numerical solver which partitions E, H and G based on the three equations. The inverse bowen ratio is purely mathematically derived. qs and Ts over canopy and soil are derived differently. See table 1. qs over canopy is derived from air relative humidity while qs over soil is derived from soil surface water potential obtained from soil moisture as detailed in the section 2.2 |
| - Lines 111–114: Comments on the flowchart in Figure 1: * This Figure describes how the various input data sets are processed by MEP to generate the desired output, evapotranspiration. Yet, none of those boxes appear to require any of the fluxes H, E or G mentioned earlier: only Rn is discussed or used subsequently. So what is the relation, if any, between the materials and equations described between lines 89 and 108, on one side, and the rest of the paper? | While the MEP method produces values for E, H and G, in this manuscript, the focus is only on E, which is the evapotranspiration, which is a sum of the evaporation and transpiration. Hence the outputs of H and G are not reported in this manuscript. The development of the MEP method has been comprehensively analysed and tested in manuscripts on point or catchment scales (Wang and Bras, 2011;Hajji et al., 2018), hence the derivations are not repeated in this manuscript, this manuscript is focused on the application of those pre-derived equations at continental scale and the development of a dataset over Australia.

Nevertheless, we will update the manuscript to explain that it is focused on the evapotranspiration component while referring the reader to the development papers to questions on the derivation of the method as a whole. |
| * Figure 1 distinguishes between "soil surface relative humidity" and "soil surface specific humidity", yet the equations or algorithms to derive those variables are nowhere to be seen... According to Equation (2), the variable soil sigma (σs) is derived from both soil surface temperature Ts and atmospheric specific humidity qs, yet, the Figure does not show this latter dependency | The equation for obtaining specific humidity from relative humidity has been included in the Appendix. The equation for sigma over target surface is derived from target surface temperature and target surface specific humidity as indicated in the Figure 1. Sigma over a target surface is dependent on the temperature and specific humidity over the target surface as indicated in Figure 1.

This will be better described in the manuscript to improve clarity |
| - Line 118: What are the source, accuracy, spatial and temporal resolutions of Rn? | The Rn was calculated from solar radiation data developed as part of the SILO data suite (Jeffrey et al., 2001). The spatial and temporal resolution will be included in the table the reviewer requested above in the updated manuscript. |
| - Line 119: FPAR is mentioned here (on page 6), but that product does not appear as an input in Figure 1. Also, it is not clear what its role is at this point (this is | The FPAR product from the MODIS satellite is used in partitioning the evaporation and transpiration component. The product is used as a surrogate for the vegetation cover in the Figure 1. This will be included in |

| | |
|---|---|
| only clarified on page 20!–see the comment on Line 292 below). | the table of input data requested by the reviewer in the updated manuscript. |
| - Line 121–125: The text appears to use the expressions "vegetation cover" and "vegetation fraction" as synonyms. This is confusing, as "vegetation cover" could be understood as "land cover", which typically refers to the type of ecosystem (forest, savanna, etc.) while "vegetation fraction" hints to the more appropriate concept of "fractional vegetation cover". This latter variable is itself generally poorly defined (and hard to estimate when the Leaf Area Index is less than 3), as is often the case in arid environments like Australia. In any case, if the fractional vegetation cover Fc is derived from FPAR, then Figure 1 should show FPAR in the crossed-out box, and Fc should appear in a separate box. | Good point, this will be updated as the fractional vegetation cover in the updated manuscript. |
| - Lines 59–61 and 131: There is an internal inconsistency in first decrying the poor quality of the MODIS temperature product and in using it nevertheless. This Reviewer cannot comment on the value of this product, but if the Authors estimate it is incorrect, they should use another one | While the authors mentioned in this line, the reported challenges with the MODIS LST, the product is still one of the best products out there in terms of spatial and temporal resolution, hence its usage in the MEP. Moreover, the authors explained the MEP is much less sensitive to the temperature input data in lines 75 – 76, which was also noted by the developer of the MEP method (Wang and Bras, 2011). We will make a note in the revised manuscript on this apparent issue. |
| - Lines 131–134: The rationale for using the lowest Ts during the month is dubious: what is the accuracy and reliability of that product, generated on an 8-day basis (according to Line 59), if the area of interest happens to be overcast on successive 8-day periods? And even if Ts is always observed at least once a month, that measurement would necessarily occur on a clear (relatively hotter) day, so there is still a bias towards high temperatures during cloudy days. What is the possible impact of that bias? | As mentioned in the comment above, the MEP is much less sensitive to temperature in the algorithm. However, as with all continental scale products, concessions would have to be made in cases such as cloud cover (there is no current perfect product without cloud cover). The authors acknowledge these limitations of producing a continental scale product, hence the comparing to results from ground-based eddy covariance data to give a degree of confidence as necessary with all such regional to continental scale products after making concessions in producing such datasets. |
| - Lines 135–149: This whole paragraph appears very confusing, because it amounts to a somewhat disparate assemblage of algorithms and equations found in the literature, using multiple empirical coefficients which may or may not be applicable to Australia. Why mention methods that are not used? And again, those tools depend on additional soil properties (whose accuracy, spatial and temporal resolutions are unknown, by the way) apparently obtained from the Australian Soil Resource Information System (ASRIS), though the latter is not reported as an input in Figure 1. As the Authors rewrite this paragraph, they would be well advised to describe the necessary steps in logical order, to | This paragraph will be revised in the updated manuscript taken the comments of the reviewer into account. |

| | |
|---|---|
| explicitly provide the full equations, to indicate clearly what inputs are needed, where they come from, their spatial and temporal resolutions, accuracy, etc. | |
| - Line 140: The text appears to indicate that the values of the soil water content at wilting point and at field capacity are fixed in space and time to -1.5MPa and -10kPa, respectively. What evidence is there that this is reasonable, given the high diversity of soil types and properties? | The text does indicate that specific soil types with specific properties have a determinable soil water content at wilting point and field capacity. It is based on this principle that the Hutson and Cass method is applied to calculate the soil surface water potential across each grid cell, based on the specific soil properties from ASRIS.

This is updated as part of the revision of the paragraph requested above |
| - Line 145: What exactly is implied here by "modest data requirement and relative accuracy"? Constant values would be even simpler... This is not an acceptable rationale: the selection of models and parameterizations should be guided by strict requirements in terms of accuracy and performance to achieve a particular objective, not in terms of data volume or approximate value. Besides, how small should a database be to be "moderate", and what is the benchmark to evaluate the "relative accuracy"? | This is updated as part of the revision of the paragraph requested above. |
| - Line 151: The authors introduce the concept of "distance z above the target surface for which the Monin-Obukhov similarity theory is valid in the formula of the thermal inertia of turbulent air above soil surface", but none of those Equations are provided. A Table of (fixed) values is provided, and a map of z over Australia is produced, but what are the significance and implications of this variable in the MEP model? Since 90% or more of that continent is covered by low vegetation anyway, does it matter to consider this parameter? | The concept of distance "z" above the target surface was not introduced by the authors in this manuscript, the concept of "z" was introduced as part of the postulation of the MEP theory of evapotranspiration and its relevance was also discussed (Wang and Bras, 2009;Wang and Bras, 2011). This paper refrained from re-deriving every parameter in the MEP equation to avoid this manuscript becoming a re-print of the MEP development paper. |
| - Line 155: What is the scientific basis for assigning those particular values in Table 1? Note also that the text refers to z as a "target distance", while the legend appears to refer to a "target surface". | Literature has been cited to support the values and the text will be updated in the manuscript to take this and the noted inconsistency into account. |
| - Line 158: In Figure 2, why is there a blue block corresponding to z values of 11 m when Table 1 does not show any land cover with that value? And why is the legend to this figure labeling z as the 'target height (z)', which would be normally understood as the height of vegetation, rather than the theoretical concept mentioned above? | This will be fixed in the manuscript along with the labelling. |
| - Line 161: The text refers to Eq. 14, but that Equation (on line 197) has nothing to do with "the Hutson and Cass coefficients a and b": this probably refers to an equation in another paper... | The equation number will be corrected in the updated manuscript. |

| | |
|---|---|
| - Line 166: Where does the soil density come from? What is its accuracy and spatial resolution? | Soil data including soil density are obtained from CSIRO see line 147-149. The resolution is 0.05 degrees, it is updated in the manuscript. |
| - Lines 171–181: The ESA CCI Soil Moisture (SM) product is delivered at the spatial resolution of 0.25◦ . Resampling it (by duplication or interpolation) on a grid at 0.05◦ may be programmatically convenient or necessary, but that does not change the intrinsic nature or properties of the product! Also, which version of that product is actually used here, since it has already been released 6 times? | The authors' aim of resampling the data was to unify the resolution of this dataset with the other input datasets and not to change the data. The version 04.4 of the data is used and it has been updated in the manuscript. |
| - Lines 186–187: Please provide an explicit description of the interpolation procedure as well as the formula to calculate the specific humidity of the atmosphere using the Clausius-Clapeyron equation: any reader should be allowed to duplicate the work without having to guess which tools, techniques or equations should be used. | The data was obtained as is from SILO and the interpolation procedure used to produce the data by the SILO team is comprehensively described in the cited published paper (Jeffrey et al., 2001). The specific humidity calculation is described in the appendix. This will be clarified in the updated manuscript. |
| - Line 200: The text mentions the variables xn and yn, but there are no such variables in the equations above, and it is not clear what distinguishes them anyway. | This is fixed in the updated manuscript. |
| - Line 201: The diacritical marks on top of the Q symbols are barely distinguishable on this line, and almost impossible to differentiate in the Equations above (Lines 193–198). Please use other signs or reformat or enlarge the Equations. | This is fixed in the updated manuscript. |
| - Lines 204–205: The wording here may be misleading: The "MEP ET product" may be technically available "at 5 km2 resolution", but to the extent the space-derived soil moisture data is the dominant input, the actual spatial resolution may be closer to 25 km by 25 km or 625 km2 , as noted in the comment above (Lines 171–181). Similarly, the statement "tower flux data with footprints ranging from 100 m2 up to about 2 km2 " is purely gratuitous, in the absence of evidence, or algorithms, to support these estimates. Hence comparing these products and claiming that this is a validation is unwarranted, on the basis of the information presented here. | Several published papers and products have been created globally with certain inputs available at lower resolutions, while other inputs are at higher resolutions, with the final resolution determined to be at the resolution of the highest data input (Jeffrey et al., 2001;Mu et al., 2013;Mark and Damien, 2015). Hence the authors are of the opinion that this is an acceptable practice in the scientific community. However, we will make clear in the revised manuscript that the spatial accuracy of the product is determined by a combination of data layers with different spatial resolutions.

The statement on the flux footprints will be revised and updated in the manuscript. |
| - Lines 229–233: Please note that there are two Figures labeled "Figure 3" on this page | This will be corrected in the updated manuscript |
| - Line 231: In the second Figure 3, what is the source of those data? What is the area concerned (the whole of Australia)? Is this the result of an accumulation of data from rain-gauges, or a satellite product, or a reanalysis? What is the accuracy of those precipitation estimates? What is the spatial representativity of the precipitation data? Do they provide a spatial coverage comparable with the gridded data of MEP? And of | The data source is the Bureau of Meteorology Australia (BOM). These are annual mean precipitation over Australia. http://www.bom.gov.au/climate/current/annual/aus/#tabs=Rainfall |

| | |
|---|---|
| course, most importantly, what is the error bar associated with the MEP estimates? | The spatial coverage is the entire Australia as the MEP. More detail on this product will be included in the revised manuscript. |
| - Line 247: The legend of Table 2, or the text (or both), should indicate that the numbers set in bold face in this table point to the best performing method to evaluate the evapotranspiration rate at the various Eddy Covariance (EC) sites, according to the criteria indicated in the table header. | This will be updated in the manuscript. |
| - Line 292: At long last, the text reveals that "The FPAR product [is] used in partitioning net radiation between soil and canopy". However, this is basically incorrect and inappropriate: FPAR is a measure of the effectiveness of the vegetation in absorbing photosynthetically active radiation (PAR), not in any way an indicator of the fractional vegetation cover. A particular value of FPAR derived from satellite observations could be obtained from a wide range of ecosystems with widely varying Fc and Leaf Area Index (LAI), not to mention other possible factors. | The NDVI product was used in the initial analysis but the FPAR was used as a surrogate for the vegetation cover due to resolution and data availability. Moreover, Los et al. (2000) after extensive analysis, opined that the spatial distribution and seasonal changes of the FPAR is in close alignment with the NDVI. Furthermore, other publications (Xu et al., 2012;Mu et al., 2011b) have successfully used the FPAR as a surrogate of the vegetation cover, hence our use of the FPAR in this study. |
| - Lines 307 and 335: Table 3 is mentioned in the text before Table 2, and no Table is labeled 3 in this document. However, there is a Table 2 starting on Line 244, and another one starting on Line 335... | This will be fixed in the updated manuscript |
| - Line 312: The Figure referred to by "as a percentage of rainfall (Fig 2)" exhibits the spatial distribution of z, not precipitations. In fact, there is no precipitation map in the paper, although one would be useful, as noted above. | This will be fixed in the updated manuscript |
| - Line 346: The statement "The MEP model appears lacking spatial continuity, probably due to the use of pedotransfer functions..." is invalid. Those functions are just mathematical formulae, fitting functions: they cannot by themselves generate spatial discontinuities. If the MEP model outputs appear spatially discontinuous, it must be because the soil moisture input data themselves are discontinuous, or because of model (coding) errors. For this reason too, it would be useful to conduct sensitivity analyses to establish to what extent each input and algorithmic parameter or equation contributes to the outcome. If the discontinuity does arise from the intrinsic variability of the soil moisture data, no amount of tuning of the pedotransfer functions will reduce those discontinuities. A contrario, if the soil moisture data are reasonably homogeneous to start with, then there may indeed be a problem with the way those functions, or other aspects of the model, are implemented. In either case, the conclusion that "Hence, further improvement to the MEP model may be achieved by improving the parameterization of the | The spatial discontinuity in the MEP has been tested and attributed to the discontinuous soil types in ASRIS when used in the model along with the pedotransfer functions to determine wilting point and field capacity. The definite boundaries of the soil types in model propagates the discontinuity through the evaporation model of the MEP. This is a necessary compromise in creating a regional model such as this. This is rightly not observed in the transpiration model which does not require the soil data. Hence the sum of the evaporation and transpiration model smoothens out the final evapotranspiration model. We will update the revised manuscript to reflect this issue better. |

| | |
|---|---|
| pedotransfer functions for each soil type." is currently unfounded. | |
| - Line 351: The text states "The low correlation of the MEP model" but does not indicate with what MEP does not correlate well. | This will be updated in the manuscript. |
| - Lines 410–411: The acknowledgment mentions the use of FLUXNET and ERAInterim reanalysis data: Why are those data sets not mentioned in the text and appear in Figure 1? | These data were only used for comparison and not in the MEP model. They were appropriately cited in Table 2. |

Hajji, I., Nadeau, D. F., Music, B., Anctil, F., and Wang, J.: Application of the Maximum Entropy Production Model of Evapotranspiration over Partially Vegetated Water-Limited Land Surfaces, Journal of Hydrometeorology, 19, 989-1005, 10.1175/jhm-d-17-0133.1, 2018.

Hu, G. C., Jia, L., and Menenti, M.: Comparison of MOD16 and LSA-SAF MSG evapotranspiration products over Europe for 2011, Remote Sensing of Environment, 156, 510-526, 10.1016/j.rse.2014.10.017, 2015.

Jeffrey, S. J., Carter, J. O., Moodie, K. B., and Beswick, A. R.: Using spatial interpolation to construct a comprehensive archive of Australian climate data, Environ Modell Softw, 16, 309-330, http://dx.doi.org/10.1016/S1364-8152(01)00008-1, 2001.

Jin, Y., Randerson, J. T., and Goulden, M. L.: Continental-scale net radiation and evapotranspiration estimated using MODIS satellite observations, Remote Sensing of Environment, 115, 2302-2319, 2011.

Los, S. O., Pollack, N. H., Parris, M. T., Collatz, G. J., Tucker, C. J., Sellers, P. J., Malmström, C. M., DeFries, R. S., Bounoua, L., and Dazlich, D. A.: A global 9-yr biophysical land surface dataset from NOAA AVHRR data, Journal of Hydrometeorology, 1, 183-199, 2000.

Mark, F., and Damien, S.-M.: MCD12C1 MODIS/Terra+Aqua Land Cover Type Yearly L3 Global 0.05Deg CMG V006 [Data set], NASA EOSDIS Land Processes DAAC, 10.5067/MODIS/MCD12C1.006, 2015.

Mu, Q., Zhao, M., and Running, S. W.: Improvements to a MODIS global terrestrial evapotranspiration algorithm, Remote Sensing of Environment, 115, 1781-1800, http://dx.doi.org/10.1016/j.rse.2011.02.019, 2011a.

Mu, Q., Zhao, M., and Running, S. W.: MODIS Global Terrestrial Evapotranspiration (ET) Product (NASA MOD16A2/A3), Algorithm Theoretical Basis Document, Collection, 5, 2013.

Mu, Q. Z., Zhao, M. S., and Running, S. W.: Improvements to a MODIS global terrestrial evapotranspiration algorithm, Remote Sensing of Environment, 115, 1781-1800, 10.1016/j.rse.2011.02.019, 2011b.

Velpuri, N. M., Senay, G. B., Singh, R. K., Bohms, S., and Verdin, J. P.: A comprehensive evaluation of two MODIS evapotranspiration products over the conterminous United States: Using point and gridded FLUXNET and water balance ET, Remote Sensing of Environment, 139, 35-49, 2013.

Wang, J., and Bras, R. L.: A model of surface heat fluxes based on the theory of maximum entropy production, Water Resour Res, 45, Artn W11422

10.1029/2009wr007900, 2009.

Wang, J., Bras, R. L., Sivandran, G., and Knox, R. G.: A simple method for the estimation of thermal inertia, Geophysical Research Letters, 37, Artn L05404

10.1029/2009gl041851, 2010.

Wang, J. F., and Bras, R. L.: A model of evapotranspiration based on the theory of maximum entropy production, Water Resour Res, 47, Artn W03521

10.1029/2010wr009392, 2011.

Wang, J. F., Bras, R. L., Nieves, V., and Deng, Y.: A model of energy budgets over water, snow, and ice surfaces, J Geophys Res-Atmos, 119, 6034-6051, 10.1002/2013jd021150, 2014.

Xu, X., Yang, D., and Sivapalan, M.: Assessing the impact of climate variability on catchment water balance and vegetation cover, Hydrol Earth Syst Sc, 16, 43, 2012.

An, R., Zhang, L., Wang, Z., Quaye-Ballard, J. A., You, J., Shen, X., Gao, W., Huang, L., Zhao, Y., and Ke, Z.: Validation of the ESA CCI soil moisture product in China, Int J Appl Earth Obs, 48, 28-36, 2016.

Dorigo, W., Wagner, W., Albergel, C., Albrecht, F., Balsamo, G., Brocca, L., Chung, D., Ertl, M., Forkel, M., Gruber, A., Haas, E., Hamer, P. D., Hirschi, M., Ikonen, J., de Jeu, R., Kidd, R., Lahoz, W., Liu, Y. Y., Miralles, D., Mistelbauer, T., Nicolai-Shaw, N., Parinussa, R., Pratola, C., Reimer, C., van der Schalie, R., Seneviratne, S. I., Smolander, T., and Lecomte, P.: ESA CCI Soil Moisture for improved Earth system understanding: State-of-the art and future directions, Remote Sensing of Environment, 203, 185-215, 10.1016/j.rse.2017.07.001, 2017.

Dorigo, W. A., Gruber, A., De Jeu, R. A. M., Wagner, W., Stacke, T., Loew, A., Albergel, C., Brocca, L., Chung, D., Parinussa, R. M., and Kidd, R.: Evaluation of the ESA CCI soil moisture product using ground-based observations, Remote Sensing of Environment, 162, 380-395, 10.1016/j.rse.2014.07.023, 2015.

McNally, A., Shukla, S., Arsenault, K. R., Wang, S., Peters-Lidard, C. D., and Verdin, J. P.: Evaluating ESA CCI soil moisture in East Africa, Int J Appl Earth Obs, 48, 96-109, 2016.

---

## Author Comment (AC2) · 1 May 2020

The authors are immensely grateful for the invaluable review, corrections and suggestions given by the reviewer.

| Reviewer comment | Author response |
|---|---|
| The results and discussion needs lots of work. It will be good to describe the spatiotemporal dynamics of MEP E and T in detail for 2003-2013 period and connect them with local and regional hydroclimatic fluctuations. Possible challenges can be expanded using uncertainty of the input dataset. What can we learn from these MEP products? What is the new knowledge this dataset can inform us that we do not know right now? What types of future studies this MEP dataset can generate for our scientific community? | The authors will include a section in the updated manuscript on the spatiotemporal dynamics of the MEP E and T for the study period however an attempt to connect them to regional hydroclimatic data was not feasible due to lack of E and T data locally and regionally. The updated discussion section will include new knowledge from the MEP and possible future studies/direction from the MEP studies. |
| Figure 3 caption is inadequate (Mean maps of which product??? MEP?). Presenting the mean annual ET is not enough to claim the conclusion. I think maps of standard deviation, ET map of max ET (the year of Maximum at study area scale) and min ET map (the year of Minimum ET at study area scale) are required | The Figure 3 caption errors will be corrected and updated while minimum and maximum ET year will also be included. |
| Figure numbering got messed up. Figure 3 was labeled twice. As a result, it was difficult to follow the manuscript. | The figures will be renumbered and corrected. |
| Line 111: What is the source of vegetation fraction (Not provided/explained?)? | The FPAR is used as a surrogate for the vegetation fraction (Los et al., 2000). This was provided in line 292. However, this will be incorporated earlier in the manuscript. |
| Figure 1: Did you use rainfall, humidity, and temperature from one station? If multiple stations, then how did you distribute/interpolate across the entire content? How much is the spatial variability of precipitation across Australia? | The MEP does not use or require rainfall data in its modelling but uses humidity and temperature. In this study, the SILO data product described and developed by Jeffrey et al. (2001) was used and cited in lines 185 and 186. This will be expanded in the updated manuscript. |
| Please add a discussion regarding the propagation of the uncertainty or errors in the input dataset to your MEP product. I believe the vegetative fraction and soil moisture have quite a bit of uncertainty. | A discussion on the uncertainty propagation by the input datasets will be included in the updated manuscript. |

| | |
|---|---|
| What is the extent of footprint for eddy covariance tower estimates? How this extent compare to the pixel size of the MEP product? Please add this discussion somewhere. | A brief on the footprint of the EC tower data will be included in the updated manuscript, however, a lack of individual EC site footprint data on the FLUXNET over Australian sites will limit this discussion. |
| Line 236-238: I don't see any ET reduction between 2003 and 2008 in figure 3. How this is consistent with Jung et al., (2010) Line 309: NSE is not reported in this manuscript. Table 2 does not show any NSE. | According to Jung et al. (2010), there was a decline in the ET over Australia between 1998 – 2008, this is evidenced in our Figure 3 which shows lower ET between 2003 – 2009, followed by a sharp increase in 2010 after the drought broke over Australia. The authors are of the opinion that this is consistent with the observation by Jung et al. (2010). The NSE between the MEP, AWRA-L and MOD16 are reported in the Table 2, with MOD16 to MEP as -0.05, AWRA-L to MEP as 0.51 and AWRA-L to MOD16 as 0.25. |

Jeffrey, S. J., Carter, J. O., Moodie, K. B., and Beswick, A. R.: Using spatial interpolation to construct a comprehensive archive of Australian climate data, Environ Modell Softw, 16, 309-330, http://dx.doi.org/10.1016/S1364-8152(01)00008-1, 2001.

Jung, M., Reichstein, M., Ciais, P., Seneviratne, S. I., Sheffield, J., Goulden, M. L., Bonan, G., Cescatti, A., Chen, J. Q., de Jeu, R., Dolman, A. J., Eugster, W., Gerten, D., Gianelle, D., Gobron, N., Heinke, J., Kimball, J., Law, B. E., Montagnani, L., Mu, Q. Z., Mueller, B., Oleson, K., Papale, D., Richardson, A. D., Roupsard, O., Running, S., Tomelleri, E., Viovy, N., Weber, U., Williams, C., Wood, E., Zaehle, S., and Zhang, K.: Recent decline in the global land evapotranspiration trend due to limited moisture supply, Nature, 467, 951-954, 10.1038/nature09396, 2010.

Los, S. O., Pollack, N. H., Parris, M. T., Collatz, G. J., Tucker, C. J., Sellers, P. J., Malmström, C. M., DeFries, R. S., Bounoua, L., and Dazlich, D. A.: A global 9-yr biophysical land surface dataset from NOAA AVHRR data, Journal of Hydrometeorology, 1, 183-199, 2000.